# The Land Variational Ensemble Data Assimilation fRamework: LaVEnDAR v1.0.0

Ewan Pinnington[1], Tristan Quaife[1,2], Amos Lawless[1,2], Karina Williams[3], Tim Arkebauer[4], and Dave Scoby[4]

[1]National Centre for Earth Observation, Department of Meteorology, University of Reading, Reading UK
[2]School of Mathematical, Physical and Computational Sciences, University Of Reading, Reading UK
[3]Met Office Hadley Centre, Exeter, UK
[4]Department of Agronomy and Horticulture, University of Nebraska-Lincoln, Lincoln, Nebraska, USA

**Correspondence:** Ewan Pinnington (e.pinnington@reading.ac.uk)

**Abstract.** The Land Variational Ensemble Data Assimilation fRamework (LaVEnDAR) implements the method of Four-Dimensional Ensemble Variational data assimilation for land surface models. Four-Dimensional Ensemble Variational data assimilation negates the often costly calculation of a model adjoint required by traditional variational techniques (such as 4DVar) for optimising parameters/state variables over a time window of observations. In this paper we present the first application of LaVEnDAR, implementing the framework with the JULES land surface model. We show the system can recover seven parameters controlling crop behaviour in a set of twin experiments. We run the same experiments at the Mead continuous maize FLUXNET site in Nebraska, USA to show the technique working with real data. We find that the system accurately captures observations of leaf area index, canopy height and gross primary productivity after assimilation and improves posterior estimates of the amount of harvestable material from the maize crop by 74%. LaVEnDAR requires no modification to the model that it is being used with and is hence able to keep up to date with model releases more easily than other DA methods.

*Copyright statement.* TEXT

## 1 Introduction

Land surface models are important tools for representing the interaction between the Earth's surface and the atmosphere for weather and climate applications. They play a key role in the translation of our knowledge of climate change into impacts on human life. Most land surface models will converge to a steady state; their state vector tends toward an equilibrium defined by forcing variables (i.e. the meteorology experienced by the model) and the model parameters. This is quite unlike fluid dynamics models used for the atmosphere and oceans, which exhibit chaotic behaviour; a small change in their initial state can lead to large deviations in the state vector evolution with time. Consequently, for some land surface applications parameter estimation can have greater utility than state estimation (Luo et al., 2015). This manuscript deals primarily with the problem of parameter

estimation in land surface models, although the technique we introduce could easily be used to for state estimation problems too.

Data Assimilation (DA) combines models and data such that resulting estimates are an optimal combination of both, taking into account all available information about respective uncertainties. DA techniques are typically derived from a Bayesian standpoint and have been largely developed to service the needs of atmospheric and ocean modelling, especially where there is a need to provide near real–time forecasts. Typically the focus of such activities is on estimating the optimal model state as the fundamental laws underlying fluid dynamics are well understood and many of the model parameters are known physical constants. However, this is not true for land surface models where parameters are much less well understood. Indeed these parameters can be allowed to change over time within a developing ecosystem or when an ecosystem is subject to a disturbance event to account for model structural inadequacies.

DA applications for land surface models are becoming increasingly common using a wide variety of techniques and estimating both state and parameters. Many studies have employed Markov chain Monte Carlo (MCMC) methods (*e.g.* Metropolis et al. (1953)) to retrieve posterior estimates of parameter and state variables (Post et al., 2018; Bloom et al., 2016; Bloom and Williams, 2015; Zobitz et al., 2014; Keenan et al., 2012; Braswell et al., 2005). These methods use a cost function to iteratively sample the posterior parameter distribution and can deal with non-Gaussian error. However MCMC methods come at a large computational cost, requiring in the order of $10^6$ model runs even for simpler models (Zobitz et al., 2011; Ziehn et al., 2012), which may be infeasible for applications at larger scales or for more complex land surface models than used in these studies. Sequential ensemble methods have also been used (such as the Ensemble Kalman Filter (EnKF) (Evensen, 2003)) in numerous studies (Kolassa et al., 2017; De Lannoy and Reichle, 2016; Quaife et al., 2008; Williams et al., 2005). These methods are relatively cheap (dependent on ensemble size) and easy to implement but for the problem of parameter estimation their sequential nature leads to retrieval of time varying parameter sets not physically consistent with the behaviour of the land surface. There is also a growing interest in model emulation, (Gómez-Dans et al., 2016; Fer et al., 2018), these techniques are extremely efficient but require some initial construction of the emulator. Another option is to use variational methods, common in numerical weather prediction. These have been shown to be an effective relatively cheap method of DA in land surface problems (Pinnington et al., 2017; Yang et al., 2016; Raoult et al., 2016; Bacour et al., 2015; Sawada and Koike, 2014; Rayner et al., 2005). However, when using gradient-based decent algorithms to minimise the variational cost function these methods require the derivative of the model code which can be costly to compute and maintain. The variational cost function can be minimised using non-gradient based optimisation routines (Pinnington et al., 2018) but comes at the cost of many more model runs to find convergence and loss of accuracy. Recently however there has been an increase in the development of new hybrid methods combining both ensemble and variational techniques (Bannister, 2016; Bocquet and Sakov, 2014; Desroziers et al., 2014; Liu et al., 2008). These methods present a way to retrieve time-invariant parameters over some time window without the need for the derivative of the model code or a debilitating number of model runs.

In this paper we present the first application of the Land Variational Ensemble Data Assimilation fRamework (LaVEnDAR) for implementing the hybrid technique of Four-Dimensional Ensemble Variational Data Assimilation (4DEnVar) with land surface models. We show LaVEnDAR applied to the Joint UK Land Environment Simulator (JULES) land surface model

(Clark et al., 2011; Best et al., 2011) with focus on the Mead continuous maize FLUXNET site Nebraska, USA (Suyker, 2016). At this site regular observations of canopy height, Leaf Area Index (LAI) and FLUXNET Gross Primary Productivity (GPP) are available.

Data assimilation has previously been implemented with the JULES land surface model with Ghent et al. (2010) using an ensemble Kalman filter to assimilate satellite observations of land surface temperature, Raoult et al. (2016) conducting experiments with Four–Dimensional Variational data assimilation focusing on the carbon cycle and Pinnington et al. (2018) assimilating satellite observations of soil moisture over Ghana. Of these studies Raoult et al. (2016) and Pinnington et al. (2018) are directly related to the technique presented here in that they used variational DA techniques to estimate parameters in JULES. Raoult et al. (2016) use an adjoint of JULES (ADJULES) in their study to estimate carbon cycle relevant parameters for different plant functional types. However the adjoint is only currently available for JULES version 2.2, and considerable effort would be required to update it to the most recent model version (5.3 as of 01/01/2019). Pinnington et al. (2018) used a more recent version of JULES (4.9) but avoided the need for an adjoint by using a Nelder–Mead Simplex algorithm to perform the cost function minimisation. This inevitably requires a greater number of model integration steps than using a derivative based technique and is unlikely to work effectively for large dimensional problems.

Our results show that 4DEnVar is a promising technique for land surface applications that is easy to implement for any land surface model and provides a reasonable trade off between the computational efficiency of a full 4DVar system and the complexity and effort of maintaining a model adjoint. Perhaps most significantly no modification to the model code itself is required. In section 2 we present the JULES model, describe the 4DEnVar technique in detail and outline the experiments conducted in the paper. Results are shown in section 3, with Discussions and Conclusions in section 4 and 5 respectively.

## 2 Method

### 2.1 JULES land surface model

The Joint UK Land Environment Simulator (JULES) is a community developed process based land surface model and forms the land surface component in the next generation UK Earth System Model (UKESM). A description of the energy and water fluxes is given in Clark et al. (2011), with carbon fluxes and vegetation dynamics described in Best et al. (2011). Current versions of JULES now include a parameterisation for crops with 4 default crop types (wheat, soy bean, maize and rice). Crop development is governed by a crop development index which increases as a function of crop-specific thermal time parameters with the crop being harvested when the development index crosses certain thresholds. The crop grows by accumulating daily NPP and partitioning this between a set of carbon pools (havestable material, leaf, root, stem, reserve), equations for JULES-crop can be found in Williams et al. (2017) appendix A1. a further description and evaluation for JULES-crop can be found in Osborne et al. (2015) and Williams et al. (2017). Williams et al. (2017) conducted a calibration and evaluation for JULES-crop at the Mead continuous maize site. The setup of JULES described in detail by Williams et al. (2017) forms the basis for the JULES runs within this paper with JULES version 4.9 being used. We drive JULES with observed meteorological forcing data of humidity, precipitation, pressure, solar radiation, temperature and wind.

## 2.2 Mead Field Observations

We have used observations from the Mead FLUXNET US-Ne1 site (Suyker, 2016) for meteorological driving and eddy co-variance carbon flux data. A description of the eddy covariance flux data and derivation of Gross Primary Productivity (GPP) is given in Verma et al. (2005). In this study we only select GPP observations corresponding to unfilled observations of Net Ecosystem Exchange (NEE) with the highest quality flag and remove zero values from outside of the growing season. It is important to note that GPP is not an observation *per se* and is derived by partitioning the net carbon flux using a model which is likely to be inconsistent with the process model we are assimilating the data into. This site has grown maize continuously since 2001 (previously the site had a 10 year history of maize-soybean rotation) on a soil of deep silty clay loam and has been the subject of many previous studies (Yang et al., 2017; Nguy-Robertson et al., 2015; Suyker and Verma, 2012; Guindin-Garcia et al., 2012; Viña et al., 2011). The site is irrigated using a center pivot system. The JULES model can be run with irrigation turned off or on, we have run the model with irrigation turned on. In addition to the FLUXNET observations there are also regular leaf area index, canopy height, harvestable material, leaf carbon and stem carbon observations. Leaf area index, harvestable material, leaf carbon and stem carbon observations are made using a method of destructive sampling and an area meter (Model LI-3100, LI-COR, Inc., Lincoln, NE) (Viña et al., 2011).

## 2.3 Data Assimilation

### 2.3.1 Four-Dimensional Variational Data Assimilation

This section follows the derivation given in Pinnington et al. (2016). In 4DVar we consider the dynamical nonlinear discretised system

$$\mathbf{z}_t = \mathbf{f}_{t-1 \to t}(\mathbf{z}_{t-1}, \mathbf{p}_{t-1}), \tag{1}$$

with $\mathbf{z}_t \in \mathbb{R}^n$ the state vector at time $t$, $\mathbf{p}_{t-1} \in \mathbb{R}^q$ the vector of $q$ model parameters at time $t-1$ and $\mathbf{f}_{t-1 \to t}$ the nonlinear model updating the state at time $t-1$ to time $t$ for $t = 1, 2, \ldots, N$. If we consider a set of fixed parameters then the value of the state at the forecast time $\mathbf{z}_t$ is uniquely determined by the initial state $\mathbf{z}_{t-1}$. As the model parameters are time-invariant their evolution is given by,

$$\mathbf{p}_t = \mathbf{p}_{t-1}, \tag{2}$$

for $t = 1, 2, \ldots, N$. We join the parameter vector $\mathbf{p}$ with the model state vector $\mathbf{z}$, giving us the augmented state vector

$$\mathbf{x} = \begin{pmatrix} \mathbf{p} \\ \mathbf{z} \end{pmatrix} \in \mathbb{R}^{q+n}. \tag{3}$$

The augmented system model is given by

$$\mathbf{x}_t = \mathbf{m}_{t-1 \to t}(\mathbf{x}_{t-1}), \tag{4}$$

where

$$\mathbf{m}_{t-1 \to t}(\mathbf{x}_{t-1}) = \begin{pmatrix} \mathbf{p}_{t-1} \\ \mathbf{f}_{t-1 \to t}(\mathbf{z}_{t-1}, \mathbf{p}_{t-1}) \end{pmatrix} = \begin{pmatrix} \mathbf{p}_t \\ \mathbf{z}_t \end{pmatrix} \in \mathbb{R}^{q+n}. \tag{5}$$

Process error could be included in equation (5) by specifying an additional term, but in this application is neglected. The vector $\mathbf{y}_t \in \mathbb{R}^{r_t}$ represents available observations at time $t$. These observations are related to the augmented state vector by the equation

$$\mathbf{y}_t = \mathbf{h}_t(\mathbf{x}_t) + \epsilon_t, \tag{6}$$

where $\mathbf{h}_t : \mathbb{R}^{q+n} \to \mathbb{R}^{r_t}$ maps the augmented state vector to the observations and $\epsilon_t \in \mathbb{R}^{r_t}$ denotes the observation errors. Often the errors $\epsilon_t$ are treated as unbiased, Gaussian and uncorrelated in time with known covariance matrices $\mathbf{R}_t$.

In 4DVar we require a prior estimate to the state and/or parameters of the system at time $0$ denoted by $\mathbf{x}^b$. This prior estimate is usually taken to have unbiased, Gaussian errors with a known covariance matrix $\mathbf{B}$. Including a prior term in 4DVar regularises the problem and ensures a locally unique solution (Tremolet, 2006). The aim of 4DVar is to find the initial state and/or parameters that minimise the distance to the prior estimate, weighted by $\mathbf{B}$, while also minimising the distance of the model trajectory to the observations, weighted by $\mathbf{R}_t$, through the set time window $0, \ldots, N$. We do this by finding the posterior augmented state that minimises the cost function

$$J(\mathbf{x}_0) = \frac{1}{2}(\mathbf{x}_0 - \mathbf{x}^b)^T \mathbf{B}^{-1}(\mathbf{x}_0 - \mathbf{x}^b) + \frac{1}{2}\sum_{t=0}^{N}(\mathbf{h}_t(\mathbf{x}_t) - \mathbf{y}_t)^T \mathbf{R}_t^{-1}(\mathbf{h}_t(\mathbf{x}_t) - \mathbf{y}_t), \tag{7}$$

$$J(\mathbf{x}_0) = \frac{1}{2}(\mathbf{x}_0 - \mathbf{x}^b)^T \mathbf{B}^{-1}(\mathbf{x}_0 - \mathbf{x}^b) + \frac{1}{2}\sum_{t=0}^{N}(\mathbf{h}_t(\mathbf{m}_{0 \to t}(\mathbf{x}_0)) - \mathbf{y}_t)^T \mathbf{R}_t^{-1}(\mathbf{h}_t(\mathbf{m}_{0 \to t}(\mathbf{x}_0)) - \mathbf{y}_t). \tag{8}$$

The state that minimises the cost function is often called the analysis or posterior estimate. The posterior estimate is found by inputting the cost function, prior estimate and the gradient of the cost function into a gradient based decent algorithm. The gradient of the cost function is given by,

$$\nabla J(\mathbf{x}_0) = \mathbf{B}^{-1}(\mathbf{x}_0 - \mathbf{x}^b) + \sum_{t=0}^{N} \mathbf{M}_{t,0}^T \mathbf{H}_t^T \mathbf{R}_t^{-1}(\mathbf{h}_t(\mathbf{x}_t) - \mathbf{y}_t) \tag{9}$$

$$\nabla J(\mathbf{x}_0) = \mathbf{B}^{-1}(\mathbf{x}_0 - \mathbf{x}^b) + \sum_{t=0}^{N} \mathbf{M}_{t,0}^T \mathbf{H}_t^T \mathbf{R}_t^{-1}(\mathbf{h}_t(\mathbf{m}_{0 \to t}(\mathbf{x}_0)) - \mathbf{y}_t) \tag{10}$$

where $\mathbf{M}_{t,0} = \mathbf{M}_{t-1}\mathbf{M}_{t-2}\cdots\mathbf{M}_0$ is the tangent linear model with $\mathbf{M}_t = \frac{\partial \mathbf{m}_{t-1 \to t}(\mathbf{x}_t)}{\partial \mathbf{x}_t}$, $\mathbf{M}_{t,0}^T$ is the model adjoint propagating the state backward in time (this is required for efficient minimisation of the cost function using gradient descent techniques) and $\mathbf{H}_t = \frac{\partial \mathbf{h}_t(\mathbf{x}_t)}{\partial \mathbf{x}_t}$ is the linearized observation operator. Both the linearized observation operator and the tangent linear model can

be difficult to compute, as discussed in section 1. In section 2.3.2 we show how 4DEnVar allows us to avoid the computation of these quantities in the gradient of the cost function. We can avoid the summation notation in the cost function and its gradient by using vector notation and rewriting as,

$$J(\mathbf{x}_0) = \frac{1}{2}(\mathbf{x}_0 - \mathbf{x}^b)^T \mathbf{B}^{-1}(\mathbf{x}_0 - \mathbf{x}^b) + \frac{1}{2}(\hat{\mathbf{h}}(\mathbf{x}_0) - \hat{\mathbf{y}})^T \hat{\mathbf{R}}^{-1}(\hat{\mathbf{h}}(\mathbf{x}_0) - \hat{\mathbf{y}}) \tag{11}$$

and

$$\nabla J(\mathbf{x}_0) = \mathbf{B}^{-1}(\mathbf{x}_0 - \mathbf{x}^b) + \hat{\mathbf{H}}^T \hat{\mathbf{R}}^{-1}(\hat{\mathbf{h}}(\mathbf{x}_0) - \hat{\mathbf{y}}), \tag{12}$$

where,

$$\hat{\mathbf{y}} = \begin{pmatrix} \mathbf{y}_0 \\ \mathbf{y}_1 \\ \vdots \\ \mathbf{y}_N \end{pmatrix}, \ \hat{\mathbf{h}}(\mathbf{x}_0) = \begin{pmatrix} \mathbf{h}_0(\mathbf{x}_0) \\ \mathbf{h}_1(\mathbf{m}_{0\to1}(\mathbf{x}_0)) \\ \vdots \\ \mathbf{h}_N(\mathbf{m}_{0\to N}(\mathbf{x}_0)) \end{pmatrix}, \ \hat{\mathbf{R}} = \begin{pmatrix} \mathbf{R}_{0,0} & \mathbf{R}_{0,1} & \dots & \mathbf{R}_{0,N} \\ \mathbf{R}_{1,0} & \mathbf{R}_{1,1} & \dots & \mathbf{R}_{1,N} \\ \vdots & \vdots & \ddots & \vdots \\ \mathbf{R}_{N,0} & \mathbf{R}_{N,1} & \dots & \mathbf{R}_{N,N} \end{pmatrix} \text{ and } \ \hat{\mathbf{H}} = \begin{pmatrix} \mathbf{H}_0 \\ \mathbf{H}_1 \mathbf{M}_0 \\ \vdots \\ \mathbf{H}_N \mathbf{M}_{N,0} \end{pmatrix}. \tag{13}$$

The matrix $\hat{\mathbf{R}}$ is a symmetric block diagonal matrix with the off-diagonal blocks representing observation error correlations in
time as discussed in Pinnington et al. (2016).

For certain applications the prior error covariance matrix $\mathbf{B}$ can become large, ill-conditioned and difficult to invert. As a result minimising the cost function in equation (11) and finding the optimised model state/parameters can be slow. To ensure the 4DVar cost function converges as efficiently as possible and to avoid the explicit computation of the matrix $\mathbf{B}$ the problem is often preconditioned using a control variable transform (Bannister, 2016). We define the preconditioning matrix $\mathbf{U}$ by,

$$\mathbf{B} = \mathbf{U}\mathbf{U}^T \tag{14}$$

and

$$\mathbf{x}_0 = \mathbf{x}^b + \mathbf{U}\mathbf{w}, \tag{15}$$

so that,

$$\mathbf{w} = \mathbf{U}^{-1}(\mathbf{x}_0 - \mathbf{x}^b). \tag{16}$$

Substituting equation (15) and (16) into the cost function (equation (11)) we find

$$J(\mathbf{w}) = \frac{1}{2}\mathbf{w}^T\mathbf{w} + \frac{1}{2}(\hat{\mathbf{h}}(\mathbf{x}^b + \mathbf{U}\mathbf{w}) - \hat{\mathbf{y}})^T \hat{\mathbf{R}}^{-1}(\hat{\mathbf{h}}(\mathbf{x}^b + \mathbf{U}\mathbf{w}) - \hat{\mathbf{y}}). \tag{17}$$

Under the tangent linear approximation that

$$\mathbf{h}_i(\mathbf{m}_{0\to i}(\mathbf{x}^b + \mathbf{U}\mathbf{w})) \approx \mathbf{h}_i(\mathbf{m}_{0\to i}(\mathbf{x}^b)) + \mathbf{H}_i \mathbf{M}_{i,0}\mathbf{U}\mathbf{w}, \tag{18}$$

we can approximate equation (17) as

$$J(\mathbf{w}) = \frac{1}{2}\mathbf{w}^T\mathbf{w} + \frac{1}{2}(\hat{\mathbf{H}}\mathbf{U}\mathbf{w} + \hat{\mathbf{h}}(\mathbf{x}^b) - \hat{\mathbf{y}})^T\hat{\mathbf{R}}^{-1}(\hat{\mathbf{H}}\mathbf{U}\mathbf{w} + \hat{\mathbf{h}}(\mathbf{x}^b) - \hat{\mathbf{y}}), \tag{19}$$

with the gradient of the cost function given as

$$\nabla J(\mathbf{w}) = \mathbf{w} - \mathbf{U}^T\hat{\mathbf{H}}^T\hat{\mathbf{R}}^{-1}(\hat{\mathbf{H}}\mathbf{U}\mathbf{w} + \hat{\mathbf{h}}(\mathbf{x}^b) - \hat{\mathbf{y}}). \tag{20}$$

As the square root of a matrix is not unique there will be multiple choices for the preconditioning matrix $\mathbf{U}$.

### 2.3.2  Four-Dimensional Ensemble Variational Data Assimilation

In this section we outline a 4DEnVar scheme using the notation defined in section 2.3.1 and following the approach of Liu et al. (2008). Given an ensemble of $N_e$ joint state-parameter vectors, we can define the perturbation matrix

$$\mathbf{X}_b' = \frac{1}{\sqrt{N_e - 1}}(\mathbf{x}^{b,1} - \overline{\mathbf{x}}^b, \mathbf{x}^{b,2} - \overline{\mathbf{x}}^b, \ldots, \mathbf{x}^{b,N_e} - \overline{\mathbf{x}}^b), \tag{21}$$

here the $N_e$ ensemble members can come from a previous forecast (in which case $\overline{\mathbf{x}}^b$ is the mean of the $N_e$ ensemble members) or from a known distribution $\mathcal{N}(\mathbf{x}^b, \mathbf{B})$ such that $\overline{\mathbf{x}}^b = \mathbf{x}^b$. Using $\mathbf{X}_b'$ we can approximate the background or prior error covariance matrix by

$$\mathbf{B} \approx \mathbf{X}_b'\mathbf{X}_b'^T. \tag{22}$$

We can then transform to ensemble space using the matrix $\mathbf{X}_b'$ as our preconditioning matrix by defining

$$\mathbf{x}_0 = \mathbf{x}_b + \mathbf{X}_b'\mathbf{w}, \tag{23}$$

where $\mathbf{w}$ is a vector of length $N_e$. Defining $\mathbf{x}_0$ in this way reduces the problem in cases where the state/parameter vector is much larger than the ensemble size ($N_e$) and also regularizes the problem in cases where the state/parameter vector contains elements of contrasting orders of magnitude. From section 2.3.1 the cost function (19) becomes

$$J(\mathbf{w}) = \frac{1}{2}\mathbf{w}^T\mathbf{w} + \frac{1}{2}(\hat{\mathbf{H}}\mathbf{X}_b'\mathbf{w} + \hat{\mathbf{h}}(\mathbf{x}^b) - \hat{\mathbf{y}})^T\hat{\mathbf{R}}^{-1}(\hat{\mathbf{H}}\mathbf{X}_b'\mathbf{w} + \hat{\mathbf{h}}(\mathbf{x}^b) - \hat{\mathbf{y}}) \tag{24}$$

with gradient

$$\nabla J(\mathbf{w}) = \mathbf{w} + \mathbf{X}_b'^T\hat{\mathbf{H}}^T\hat{\mathbf{R}}^{-1}(\hat{\mathbf{H}}\mathbf{X}_b'\mathbf{w} + \hat{\mathbf{h}}(\mathbf{x}^b) + \hat{\mathbf{y}}). \tag{25}$$

We can see that the tangent linear model and adjoint are still present in equation (24) and (25) within $\hat{\mathbf{H}}$ (see equation (13)). However, we can write $\mathbf{X}_b'^T\hat{\mathbf{H}}^T$ as $(\hat{\mathbf{H}}\mathbf{X}_b')^T$ where $\hat{\mathbf{H}}\mathbf{X}_b'$ is a perturbation matrix in observation space given by

$$\hat{\mathbf{H}}\mathbf{X}_b' \approx \frac{1}{\sqrt{N_e - 1}}(\hat{\mathbf{h}}(\mathbf{x}^{b,1}) - \hat{\mathbf{h}}(\overline{\mathbf{x}}^b), \hat{\mathbf{h}}(\mathbf{x}^{b,2}) - \hat{\mathbf{h}}(\overline{\mathbf{x}}^b), \ldots, \hat{\mathbf{h}}(\mathbf{x}^{b,N_e}) - \hat{\mathbf{h}}(\overline{\mathbf{x}}^b)), \tag{26}$$

the gradient then becomes

$$\nabla J(\mathbf{w}) = \mathbf{w} + (\hat{\mathbf{H}}\mathbf{X}'_b)^T \hat{\mathbf{R}}^{-1}(\hat{\mathbf{H}}\mathbf{X}'_b \mathbf{w} + \hat{\mathbf{h}}(\mathbf{x}^b) - \hat{\mathbf{y}}), \tag{27}$$

avoiding the computation of the tangent linear and adjoint models as we can calculate (26) using only the nonlinear model and nonlinear observation operator.

### 2.3.3 Implementation with JULES

In order to implement 4DEnVar we construct an ensemble of parameter vectors and then run the process model for each unique parameter vector over some predetermined time window. We then extract the ensemble of model-predicted observations from then ensemble of model runs and compare these with the observations to be assimilated over the given time window. In our code (Pinnington, 2019) we implement the method of 4DEnVar with JULES using a set of Python modules. The data assimilation routines and minimization are included in `fourdenvar.py`. This part of the code does not need to be modified to be used with a new model. Model specific routines for running JULES are found in `jules.py` and `run_jules.py`. JULES is written in FORTRAN with its parameters being set by FORTRAN namelist (NML) files, `jules.py` and `run_jules.py` operate on these NML files updating the parameters chosen for optimisation. The data assimilation experiment is setup in `experiment_setup.py` with variables set for output directories, model parameters, ensemble size and functions to extract observations for assimilation. The module `run_experiment.py` runs the ensemble of model runs and executes the experiment as defined by `experiment_setup.py`. Some experiment specific plotting routines are also included in `plot.py`. More information and a tutorial can be found at https://github.com/pyearthsci/lavendar.

To use another model in this framework new wrappers would have to be written to mimic the functionality of `jules.py` and `run_jules.py` and allow for multiple model runs to be conducted while varying parameters. The module `run_experiment.py` would need to be updated to account for these new wrappers and functions to extract the observations for assimilation included in `experiment_setup.py`. Although we have used Python here to implement a stand alone setup of LaVEnDAR we envisage that the technique could be added to existing workflow systems such as Cylc (Oliver et al., 2019) or the Predictive Ecosystem Analyzer (PEcAn) (LeBauer et al., 2013).

### 2.3.4 Tests of the Four-Dimensional Ensemble Variational Data Assimilation System

It is important to ensure correctness of 4DEnVar the system. We show that our system is correct and passes tests for the gradient of the cost function (Li et al., 1994; Navon et al., 1992). For the cost function $J$ and its gradient $\nabla J$ we show that our implementation of $\nabla J$ is correct using the identity,

$$f(\eta) = \frac{|J(\mathbf{w} + \eta\mathbf{b}) - J(\mathbf{w})|}{\alpha \mathbf{b}^T \nabla J(\mathbf{w})} = 1 + O(\eta), \tag{28}$$

where $\mathbf{b}$ is a vector of unit length and $\eta$ is a parameter controlling the size of $\mathbf{b}$. For small values of $\eta$ we should find $f(\eta)$ close to 1. Figure 1 shows $|f(\eta) - 1|$ for a year's assimilation window with $\mathbf{b} = \mathbf{w}||\mathbf{w}||^{-1}$ where $\mathbf{w}$ is calculated from the prior

parameter values (see table 3) perturbed by 30%. We can see that $|f(\eta) - 1| \to 0$ as $\eta \to 0$ as expected, until $f(\eta)$ gets too close to machine precision at $O(\eta) = 10^{-9}$. This was also tested with different choices of **b** finding similar results.

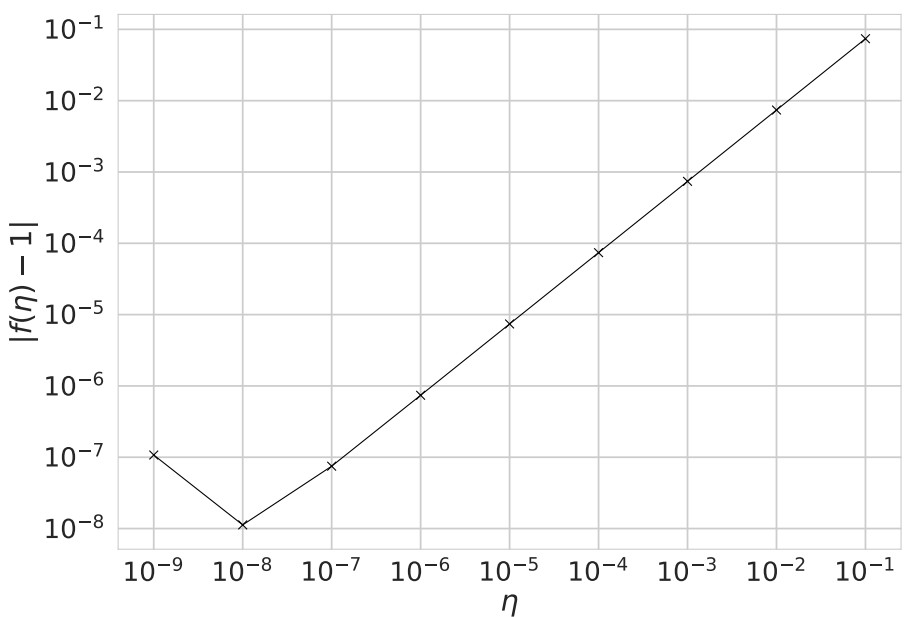

**Figure 1.** Test of the gradient of the 4DEnVar cost function.

## 2.4 Experiments

### 2.4.1 Twin Experiments

A so–called "twin" experiment in data assimilation is one where a model is used to generate synthetic observations to be assimilated. This is a commonly used approach to test whether particular combinations of observations can, in principle, be used retrieve desired target variables using some DA method. In effect the model the observations are being assimilated into is "perfect" because it represents the underlying physics that gave rise to them in the first place. We conducted a parameter estimation twin experiment with the aim to recover values for key JULES–Crop parameters: the quantum efficiency of photosynthesis,

nitrogen use efficiency (scale factor relating Vcmax with leaf nitrogen concentration), scale factor for dark respiration, two allometric coefficients for calculation of senescence and two coefficients for determining specific leaf area (see table 1). These seven parameters have an effect on the crop's seasonal growth cycle and its photosynthetic response to meteorological forcing data. The choice of parameters was motivated by the analysis of Williams et al. (2017) who found that they were least able to constrain these parameters with the available data. We assimilated synthetic observations of Gross Primary Productivity (GPP),

Leaf Area Index (LAI) and canopy height, all generated by JULES, over a year long assimilation window.

The model truth was taken from the values given in Williams et al. (2017) and perturbed using a normal distribution with a 10% standard deviation to find a prior parameter vector, $\mathbf{x}_b$. We then generated an ensemble by drawing 50 parameter vectors from the normal distribution with mean $\mathbf{x}_b$ and variance $(0.15 \times \mathbf{x}_b)^2$. Synthetic observations were sampled from the model truth with the same frequency as the real observations available from Mead and perturbed using Gaussian noise with a standard deviation of 2% of the synthetic truth value. This provided an idealised test case where we have high confidence in the assimilated observations to ensure our system is working and can recover a set of known parameters, given known prior and observation error statistics. We also include a twin experiment using the same error statistics as those used for the real data experiments at the Mead site (outlined in section 2.4.2) in supplementary material section S1.1.

| Parameter | Description | $\mathbf{x}_{true}$ |
|---|---|---|
| $\alpha$ | quantum efficiency of photosynthesis (mol $CO_2$ mol$^{-1}$ PAR) | 0.055 |
| $n_{eff}$ | nitrogen use efficiency (mol $CO_2$ m$^{-2}$ s$^{-1}$ kg C (kg N)$^{-1}$) | $5.7 \times 10^{-4}$ |
| $f_d$ | scale factor for dark respiration (-) | 0.0096 |
| $\mu$ | allometric coefficient for calculation of senescence (-) | 0.02 |
| $\nu$ | allometric coefficient for calculation of senescence (-) | 4.0 |
| $\gamma$ | coefficient for determining specific leaf area (-) | 17.6 |
| $\delta$ | coefficient for determining specific leaf area (-) | -0.33 |

**Table 1.** Description of parameters optimised in experiments and model truth value.

### 2.4.2 Mead Experiments

For the experiments using real data from the Mead US-Ne1 FLUXNET site the same seven parameters were optimised (shown in table 1) by assimilating observations over a year long assimilation window in 2008. The prior parameter vector, $\mathbf{x}_b$, is taken from the values given in Williams et al. (2017). We then generated an ensemble of 50 parameter vectors by sampling from the normal distribution with mean $\mathbf{x}_b$ and variance $(0.25 \times \mathbf{x}_b)^2$. We apply the same variance to all parameters here as the analysis of Williams et al. (2017) showed these parameters to all be poorly constrained with the available data in a more traditional model calibration study. In reality it is unlikely that all parameters will have the same variance but in the absence of additional information and for the purposes of this demonstration we used $(0.25 \times \mathbf{x}_b)^2$. Observations for the site are described in section 2.2. We prescribe a 5% standard deviation for canopy height and leaf area index errors and a 10% standard deviation for errors in GPP. These uncertainties are rough estimates that we considered adequate for demonstrating our system, but for any specific application the errors estimates should be determined more carefully. However, our uncertainties are consistent with Schaefer et al. (2012) who found an uncertainty of 1.04 g C m$^{-2}$ day$^{-1}$ to 4.15 g C m$^{-2}$ day$^{-1}$ (scaling with flux magnitude) for estimates of GPP, Raj et al. (2016) who found an uncertainty in the order of 10% for daily estimates of GPP and Guindin-Garcia et al. (2012) who found a standard error of 0.15 m$^2$ m$^{-2}$ for destructively sampled green LAI at the Mead flux site. The error statistics used within the data assimilation experiments could be investigated more thoroughly but are

appropriate for demonstrating the validity of the technique and providing an optimal weighting between prior and observation estimates.

## 3 Results

### 3.1 Twin Experiments

Figure 2 to 4 show plots of the 3 target variables over the year long assimilation window. For these figures the blue line and shading represents the 50 member prior ensemble mean and spread ($+/-$ 1 standard deviation), the orange line and corresponding shading represent the same but for the 50 member posterior ensemble of JULES model runs, pink dots with vertical lines are the synthetic observations with error bars ($+/-$ 1 standard deviation) and the dashed black line is the trajectory of the JULES model using the "true" parameter values. Figure 2 shows that after data assimilation the posterior model estimate

tracks the model truth trajectory closely with the LAI model truth always being captured by the posterior ensemble spread. For GPP Figure 3 shows a very similar result as for LAI with the posterior estimate fully capturing the model truth. Figure 4 illustrates the effect the large spread of the prior ensemble has on harvest dates towards the end of the season, with the ensemble spread increasing markedly as different ensemble members are harvested on different days. The spread for the posterior estimate of canopy height reduces considerably and tracks the model truth well. Figure 5 shows prior, posterior and

true trajectories for harvestable material. We have not assimilated any observations of this quantity but this Figure shows we improve predictions of harvestable material after assimilation of the 3 previously discussed target variables. In table 2 we show Root-Mean Squared Error (RMSE) for the 3 target variables before and after assimilation. We find an average 93.67% reduction in RMSE for the 3 target variables.

  Prior and posterior distributions for the seven parameters are shown in Figure 6 (light grey and dark grey respectively) with

20 the true model parameter values shown as dashed black vertical lines. For all seven parameters the posterior distribution moves toward the model truth and in most cases the posterior distribution mean appears very close to the model truth. The posterior distributions also narrow significantly in comparison to the prior distributions with the exception of $f_d$. Table 3 shows the mean prior and posterior parameter vectors and percentage error values between prior parameter estimates and the model truth and posterior parameter estimates and the model truth. The percentage error in the posterior estimate is reduced for all parameters,

again with the exception of $f_d$. The inability of the technique to recover $f_d$ is discussed further in section 4.1. There is an average error of 10.32% in the prior parameter estimates and this is reduced to 2.93% for the posterior estimates.

| Target variable | $\mathbf{x}_b$ RMSE | $\mathbf{x}_a$ RMSE |
|:---:|:---:|:---:|
| LAI | 1.95 | 0.15 |
| GPP | 5.17 | 0.33 |
| Canopy height | 0.39 | 0.03 |

**Table 2.** 4DEnVar twin assimilated observation RMSE for the 4 target variables when an ensemble of size 50 is used in experiments.

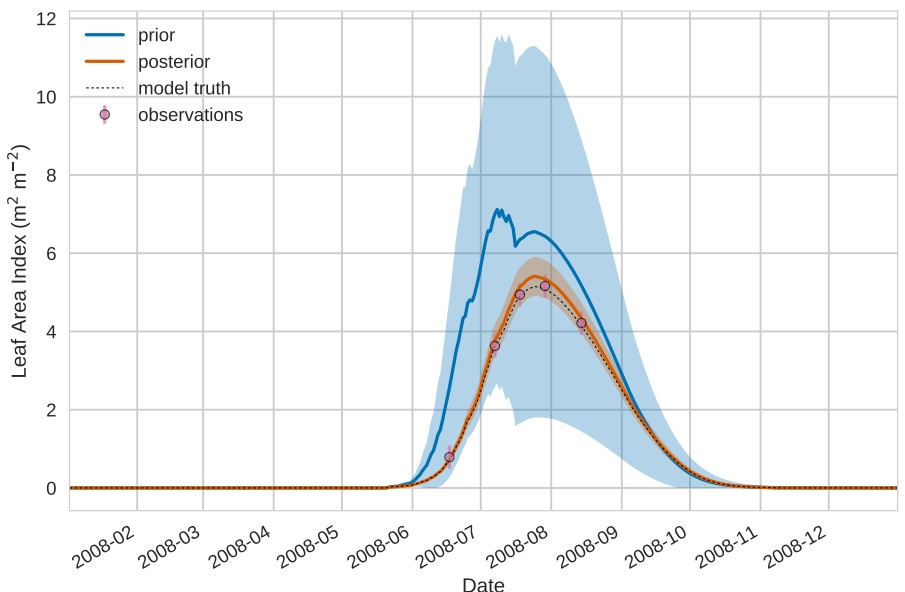

**Figure 2.** 4DEnVar twin results for leaf area index using 50 ensemble members. Blue shading: prior ensemble spread ($+/- 1\ \sigma$), orange shading: posterior ensemble spread ($+/- 1\ \sigma$), pink dots: observations with error bars, dashed black line: model truth.

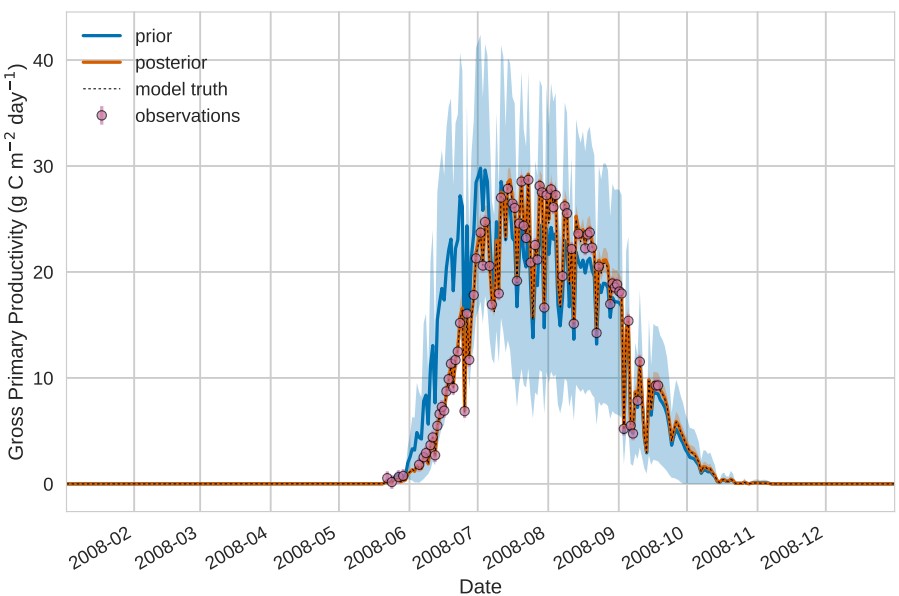

**Figure 3.** 4DEnVar twin results for gross primary productivity using 50 ensemble members. Blue shading: prior ensemble spread ($+/- 1\ \sigma$), orange shading: posterior ensemble spread ($+/- 1\ \sigma$), pink dots: observations with error bars, dashed black line: model truth.

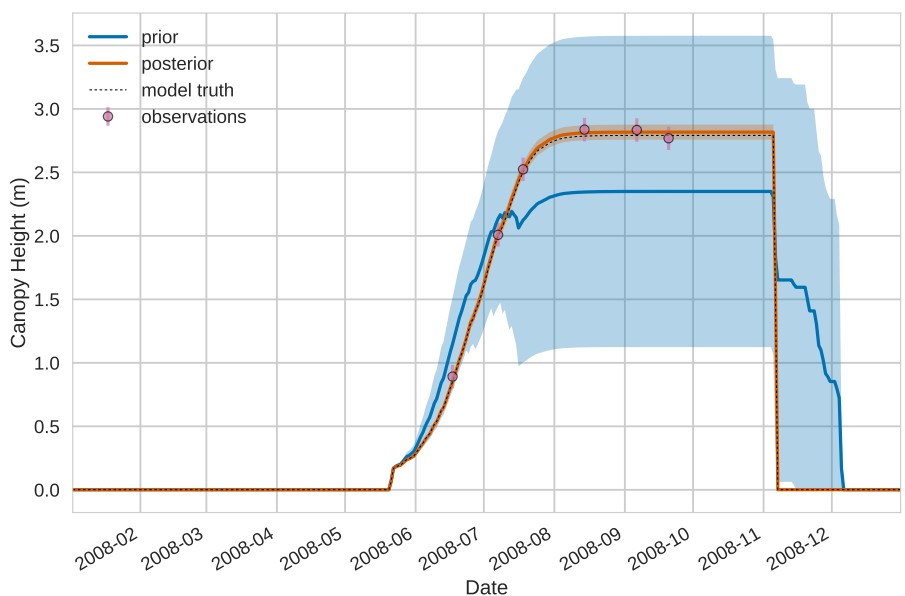

**Figure 4.** 4DEnVar twin results for canopy height using 50 ensemble members. Blue shading: prior ensemble spread ($+/- 1\ \sigma$), orange shading: posterior ensemble spread ($+/- 1\ \sigma$), pink dots: observations with error bars, dash black line: model truth.

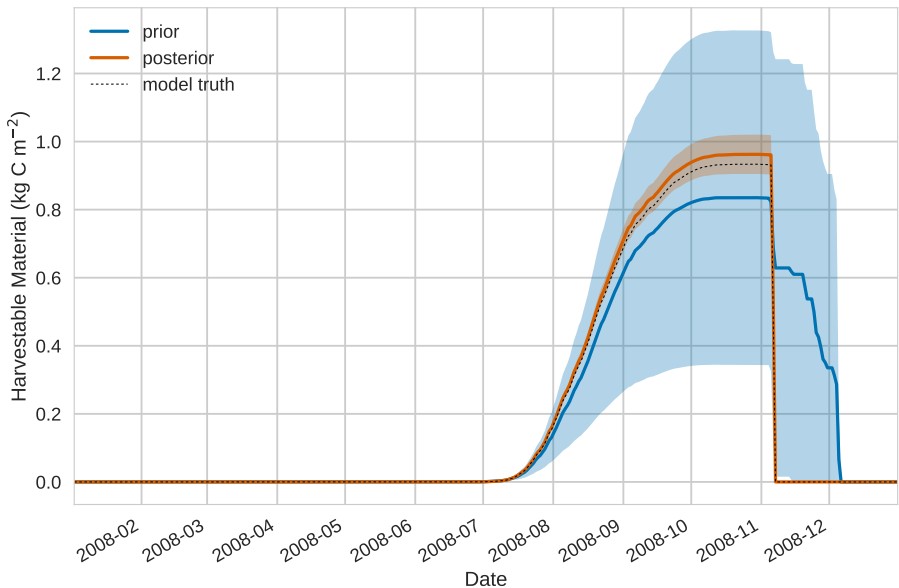

**Figure 5.** 4DEnVar twin results for harvestable material using 50 ensemble members. Blue shading: prior ensemble spread ($+/- 1\ \sigma$), orange shading: posterior ensemble spread ($+/- 1\ \sigma$), pink dots: observations with error bars, dashed black line: model truth.

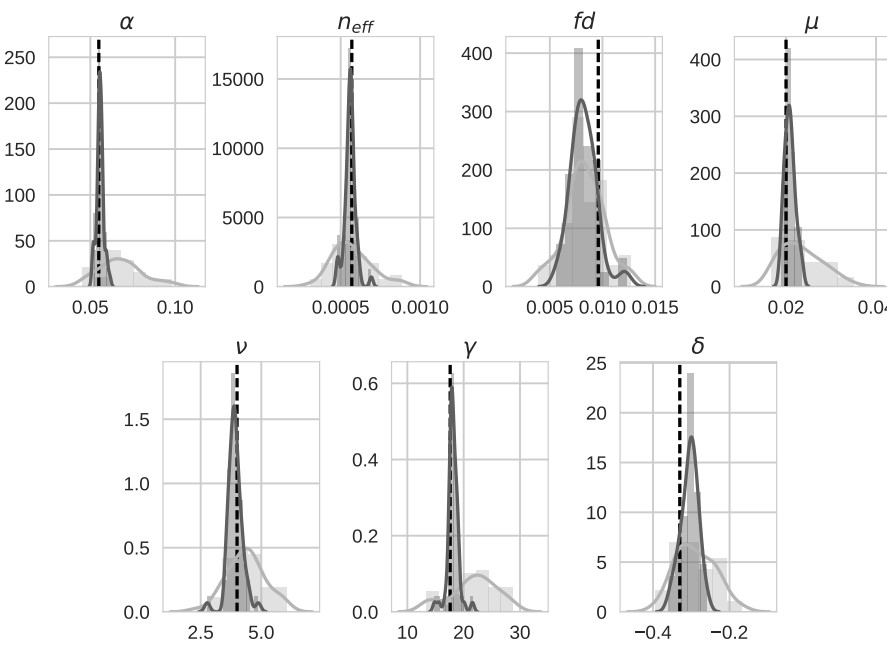

**Figure 6.** 4DEnVar twin distributions for the 7 optimised parameters for both the prior ensemble (light grey) and posterior ensemble (dark grey). The value of the model truth is shown as a dashed vertical black line.

| parameter | $\mathbf{x}_{true}$ | $\mathbf{x}_b$ | $\mathbf{x}_a$ | $\mathbf{x}_b$ % error | $\mathbf{x}_a$ % error |
|:---:|:---:|:---:|:---:|:---:|:---:|
| $\alpha$ | 0.055 | 0.067 | 0.056 | 22.4 | 1.1 |
| $n_{eff}$ | 0.00057 | 0.00062 | 0.00056 | 9.5 | 2.2 |
| $f_d$ | 0.0096 | 0.0087 | 0.0082 | 9.8 | 14.6 |
| $\mu$ | 0.020 | 0.024 | 0.021 | 18.7 | 5.3 |
| $\nu$ | 4.0 | 4.16 | 3.90 | 4.0 | 2.4 |
| $\gamma$ | 17.6 | 20.7 | 18.1 | 17.6 | 2.9 |
| $\delta$ | -0.33 | -0.29 | -0.30 | 9.8 | 8.0 |

**Table 3.** 4DEnVar twin results and percentage error for each of the seven optimised parameters when an ensemble of size 50 is used in experiments.

## 3.2 Mead field observations

Figures 7 to 9 show assimilation results for the 3 target variables over the year long window for the Mead field site. For these Figures the blue line and shading represents the 50 member prior ensemble mean (taken from Williams et al. (2017)) and spread ($+/-$ 1 standard deviation), the orange line and shading represents the same but for the 50 member posterior ensemble of JULES model runs after data assimilation and the pink dots with vertical lines are the field observations from Mead site US-Ne1 with error bars ($+/-$ 1 standard deviation). From Figure 7 we can see that the prior mean underestimates LAI, reaching a much lower peak than observations, despite this the technique finds a posterior mean estimate that agrees well with all but 2 LAI observations (in September and October). We find similar results for GPP in Figure 8, with the posterior capturing the majority of observations but missing some of the highest values. For canopy height in Figure 9 the effect of the spread in ensemble harvest dates for the prior is again obvious (also seen in the twin experiments, Figure 4), this spread is reduced for the posterior estimate and all observations are captured by the posterior ensemble spread.

Prior and posterior estimates for unassimilated independent observations are shown in Figure 10 to 12. From Figure 10 we can see the prior estimate is underestimating the amount of harvestable material for the maize crop. After assimilation the posterior estimate predicts the amount of harvestable material well and with increased confidence. Figure 11 shows that our posterior estimate of leaf carbon content improves after assimilation but is still too low, this is the same for stem carbon content in Figure 12. The fact that we can find good agreement for LAI with a poorer fit to leaf carbon content is likely due to the optimised parameters controlling specific leaf area compensating for errors in model parameters controlling the partitioning of net primary productivity into the leaf carbon pool. This allows us to achieve the correct leaf area with the incorrect leaf carbon content.

Prior and posterior ensemble parameter distributions are shown in Figure 13. After assimilation the distributions have shifted and narrowed for all parameters, except $f_d$, with $\alpha$ being the most extreme example of this. The effect these updated parameter distributions have on the model prediction of the 3 target variables in table 4 is clear. We find the largest reduction in RMSE for canopy height (73%) with the smallest reduction in RMSE for GPP (44%), overall we found an average 59% reduction in RMSE for the 3 target variables. From table 5 we can see the updated parameters have also reduced the model prediction RMSE in independent unassimilated observations. The largest reduction is in the prediction of harvestable material (74%), overall we have found an average 47% reduction in RMSE for the 3 independent observation types.

| Target variable | $\mathbf{x}_b$ RMSE | $\mathbf{x}_a$ RMSE | Reduction |
|:---:|:---:|:---:|:---:|
| LAI | 1.49 | 0.60 | 59% |
| GPP | 3.86 | 2.15 | 44% |
| Canopy height | 0.38 | 0.10 | 73% |

**Table 4.** 4DEnVar Mead assimilated observation RMSE for the 3 target variables when an ensemble of size 50 is used in experiments.

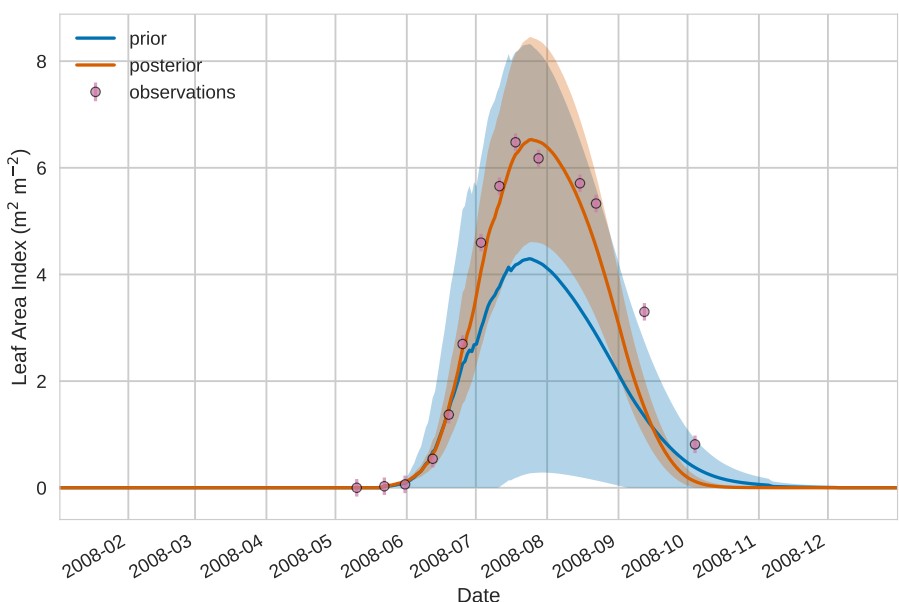

**Figure 7.** 4DEnVar results for leaf area index using 50 ensemble members. Blue shading: prior ensemble spread ($+/- 1\ \sigma$), orange shading: posterior ensemble spread ($+/- 1\ \sigma$), pink dots: observations with error bars.

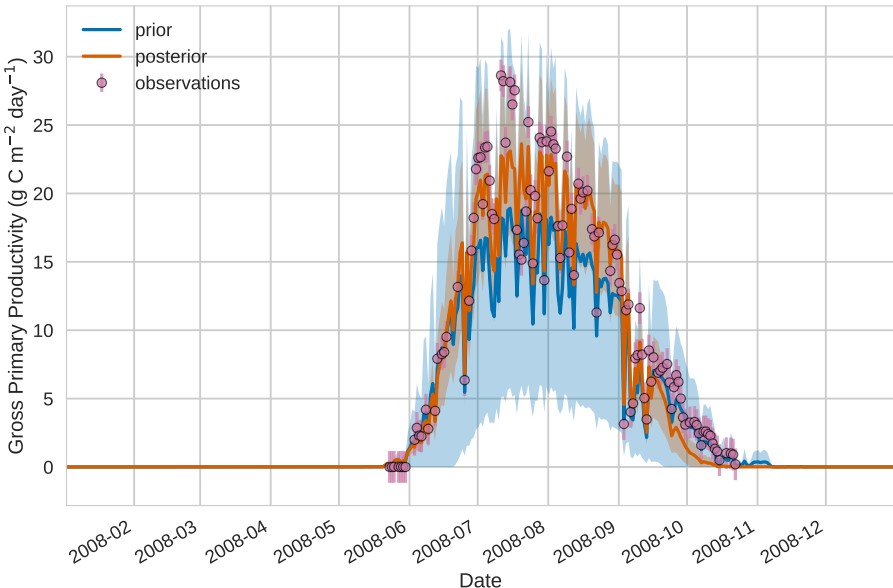

**Figure 8.** 4DEnVar results for gross primary productivity using 50 ensemble members. Blue shading: prior ensemble spread ($+/- 1\ \sigma$), orange shading: posterior ensemble spread ($+/- 1\ \sigma$), pink dots: observations with error bars.

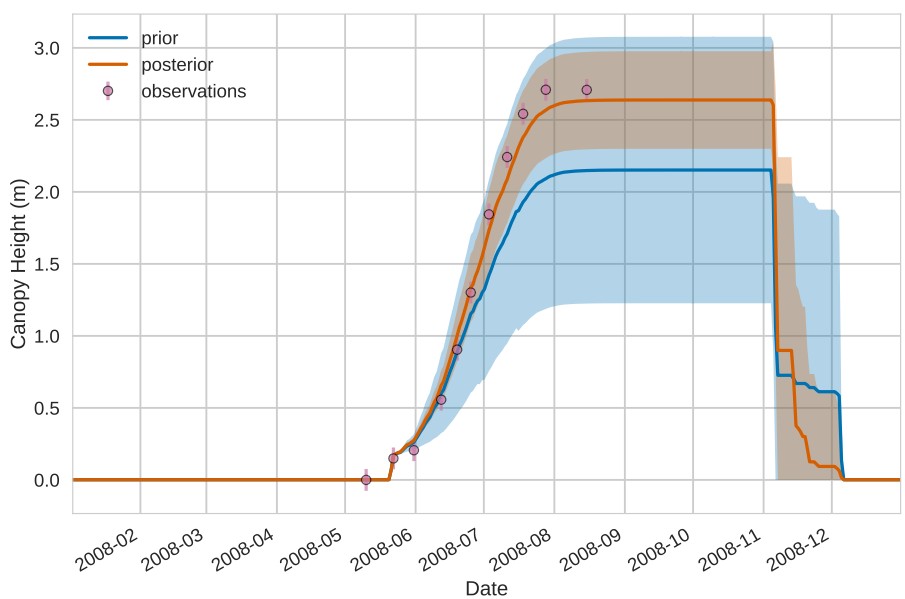

**Figure 9.** 4DEnVar results for canopy height using 50 ensemble members. Blue shading: prior ensemble spread $(+/- 1\ \sigma)$, orange shading: posterior ensemble spread $(+/- 1\ \sigma)$, pink dots: observations with error bars.

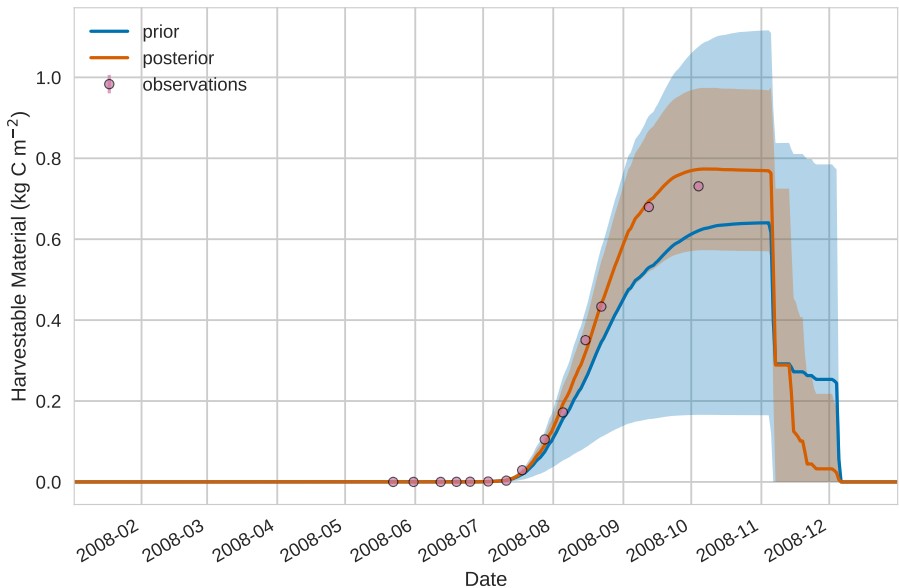

**Figure 10.** 4DEnVar results for harvestable material using 50 ensemble members. Blue shading: prior ensemble spread $(+/- 1\ \sigma)$, orange shading: posterior ensemble spread $(+/- 1\ \sigma)$, pink dots: observations.

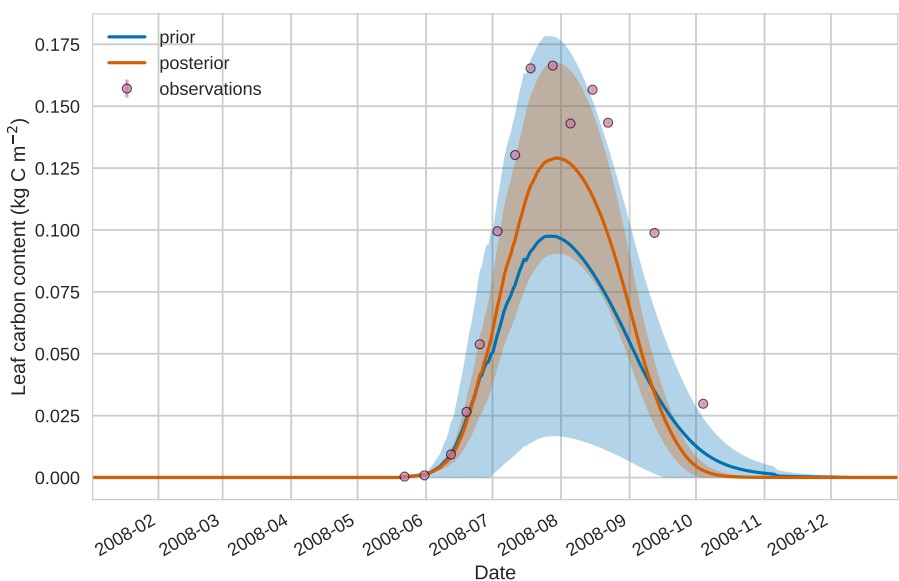

**Figure 11.** 4DEnVar results for leaf carbon using 50 ensemble members. Blue shading: prior ensemble spread ($+/- 1\ \sigma$), orange shading: posterior ensemble spread ($+/- 1\ \sigma$), pink dots: observations.

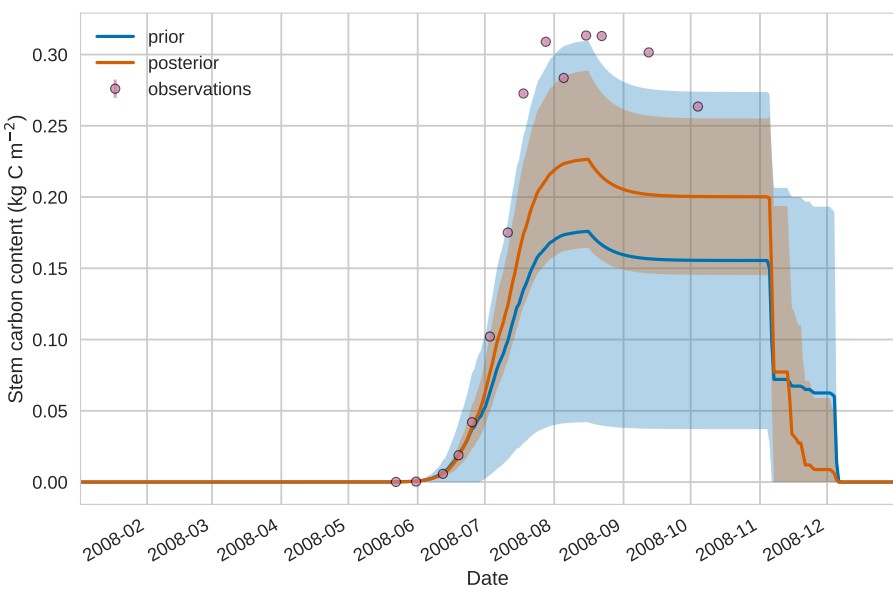

**Figure 12.** 4DEnVar results for stem carbon using 50 ensemble members. Blue shading: prior ensemble spread ($+/- 1\ \sigma$), orange shading: posterior ensemble spread ($+/- 1\ \sigma$), pink dots: observations.

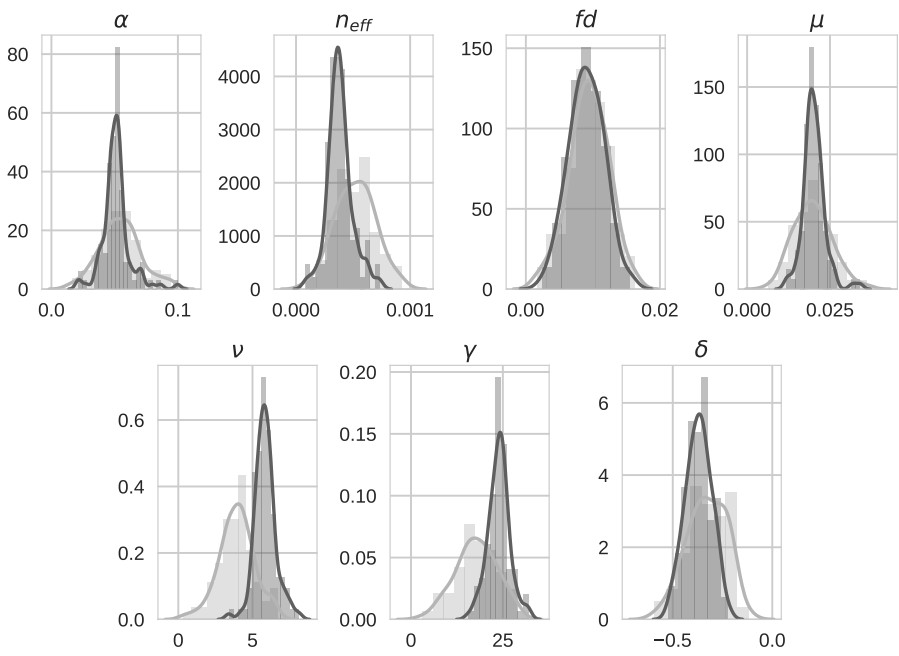

**Figure 13.** 4DEnVar distributions for the 7 optimised parameters for both the prior ensemble (light grey) and posterior ensemble (dark grey).

| Target variable | $\mathbf{x}_b$ RMSE | $\mathbf{x}_a$ RMSE | Reduction |
|---|---|---|---|
| Harvestable material | 0.06 | 0.02 | 74% |
| Leaf carbon | 0.05 | 0.03 | 32% |
| Stem carbon | 0.10 | 0.06 | 34% |

**Table 5.** 4DEnVar Mead unassimilated observation RMSE when an ensemble of size 50 is used in experiments.

## 4    Discussion

### 4.1    Twin experiments

In section 3.1 we have demonstrated that the 4DEnVar technique is able to retrieve a synthetic truth given known prior and observation error statistics. There is good agreement between the mean posterior trajectory and model truth for the 3 target

5    variables (see Figure 2, 3 and 4). We also retrieve accurate predictions of independent unobserved quantities such as harvestable material (see Figure 5). The mean posterior parameter vector after assimilation is very close to the model truth as shown in table 3 and Figure 13 with the exception of the scale factor for dark respiration $f_d$. Our inability to recover this parameter is likely due to the fact that the assimilated daily averaged observations are not greatly impacted by changes in dark respiration. Assimilating total above ground carbon could improve the estimation of $f_d$ by giving us a proxy to the net primary productivity

of the crop and with the concurrent assimilation of GPP better constraint on respiration. Alternatively including correlations in the prior error covariance matrix would provide information to update $f_d$ even when the assimilated observations are not impacted by changes in this parameter. It has been shown that suitable correlations can be diagnosed by sampling from a set of predetermined ecological dynamical constraints and taking the covariance of an ensemble run forward over a set time window (Pinnington et al., 2016).

In the results for all predicted variables we find that the posterior ensemble converges around the model truth. This can be also be seen for the parameters in Figure 6 where the posterior ensemble spread of the parameter $\alpha$ is particularly narrow. This could lead to problems when using our posterior estimate as the prior for a new assimilation cycle. It is also possible that equifinality could become an issue when attempting to optimise a larger number of parameters. From table 3 we can see this issue for the two parameters controlling photosynthetic response with the posterior slightly over-predicting $\alpha$ and under-predicting $n_{eff}$, as different combinations of these parameters can produce the same trajectory for the observed target variables. The effect of equifinality can be seen more clearly for the posterior ensemble correlation matrix included in Figure S7 of the supplementary material. It is also clear that selection of the prior ensemble is important to the success of the technique. From Figures 4 and 5 it can be seen that the prior ensemble is poor, suggesting that it could be better conditioned to deal with the discontinuity of the harvest date. It may be the case that for more complex problems an iterative step in the assimilation would be needed to address this (Bocquet, 2015) or ensemble localisation in time. In this study we have only considered the uncertainty in the parameters and initial conditions and not the uncertainty in forcing data, random effects (parameter variability) or uncertainty in the process model (Dietze, 2017). The inclusion of these additional sources of error would avoid the ensemble converging too tightly around any given value. In order to include uncertainty in the forcing data it would be necessary to run each ensemble member with a different realisation of the driving meteorology. Process error could be included in equation (5) resulting in a new term in the 4DEnVar cost function in equation (24) containing a model error covariance matrix, it has also been shown that these different types of uncertainty could be built into the observation error covariance matrix **R** (Howes et al., 2017). If estimates to these sources of error are not available the use of methods such as ensemble inflation (Anderson and Anderson, 1999), a set of techniques where the ensemble spread is artificially inflated, will help alleviate problems of ensemble convergence.

### 4.2 Mead field observations

We have demonstrated the ability of the technique to improve JULES model predictions using real data in section 3.2. Posterior estimates improve the fit to observations with the posterior ensemble spread capturing the majority of assimilated observations (see Figure 7, 8 and 9). We reduce the RMSE in the mean model prediction by an average of 59% for the 3 target variables. As independent validation that we are improving the skill of the JULES model we also improve the fit to three unassimilated observation types (see Figure 10, 11 and 12) with an average reduction in RMSE of 47%. We find the largest reduction in RMSE for the independent observations for harvestable material (74% reduction) which is an important variable closely linked to crop yield. The improvement in skill for the unassimilated observations gives us confidence that the technique has updated the model parameters in a physically realistic way and we have not over-fitted to the assimilated data. By conducting a hindcast

for 2009 (shown in supplementary material Figure S6 and table S2) we also find the retrieved posterior ensemble improves the fit to the unassimilated observations in the subsequent year, with an average reduction in RMSE of 54% when compared with the prior estimate.

The experiments with Mead field observations do not show the same level of reduction in ensemble spread as in the twin experiments (see Figure 13) due to the specified prior and observations errors being much larger. However, the posterior distribution for some parameters is still quite narrow. We again find very little update for $f_d$ as in the twin experiments, suggesting that the assimilated observations (at their current temporal resolution) are not sensitive to changes in this parameter. In our experiments we have held back observations of harvestable material, leaf carbon and stem carbon to use as independent validation of the technique. However, these observations could have been included in the assimilation to better constrain the current parameters or consider a larger parameter set.

## 4.3 Challenges and opportunities

Avoiding the computation of an adjoint makes the technique of 4DEnVar much easier to implement and also agnostic about the land surface model used. By maintaining a variational approach and optimising parameters over a time window against all available observations we also avoid retrieving non-physical time-varying parameters associated with more common sequential ensemble methods. However, as with other ensemble techniques results are dependent on having a well conditioned prior ensemble. Methods of ensemble localisation (Hamill et al., 2001), where distant correlations or ensemble members are down-weighted or removed, could be used to improve prior estimates. In this instance we would need to consider localisation in time (Bocquet, 2015). In order to extend this framework to model runs over a spatial grid we will need a method to sample prior parameter distributions regionally or globally, it would then be possible to conduct parameter estimation experiments over a region, either on a point by point basis or for the whole area at once. Considering a large area would increase the parameter space and require more ensemble members. Localisation in space could help to reduce the parameter space and thus allow for use of a smaller ensemble. The ensemble aspect of the technique also allows us to retrieve posterior distributions of parameters whereas in pure variational methods we would only find a posterior mean. However, this also presents a possible issue of posterior ensemble convergence around certain parameters. Including additional sources of error within the assimilation system (driving data error, parameter variability, process error) or using methods such as inflation (Anderson and Anderson, 1999) will help to avoid this and ensure our posterior estimates maintain enough spread to be used as a prior estimate in new assimilation cycles. While posterior parameter estimates could be used in future studies with their associated uncertainties we envisage that cycling of the assimilation system will be more appropriate for state estimation (after initial parameter estimation) where the system could be cycled on a timescale suitable for the required state variable and data availability.

In 4DEnVar we approximate the tangent linear model using an ensemble perturbation matrix. Without the explicit knowledge of the tangent linear and adjoint models 4DEnVar could be less able to deal with nonlinearities in the process model in cases where the ensemble is small or ill-conditioned. For the examples presented in this paper 4DEnVar deals well with the nonlinearity of the JULES land surface model. However, it is possible that for high dimensional spaces a technique of stochastic ensemble iteration (Bocquet and Sakov, 2013) will need to be implemented to cope with increased nonlinearity at the cost of

multiple model runs within the minimisation routine. The framework proposed in this paper allows for the implementation of such a technique fairly easily.

In this paper we have focused on using LaVEnDAR for parameter estimation. However, the technique we present can just as easily be used to adjust the model state at the start of an assimilation window in much the same way as is done in weather forecasting (Liu et al., 2008). In this case it is likely that a shorter assimilation window would be required. The posterior ensemble is then used to provide the initial conditions for the next assimilation window. This would require additional modules to be written within LaVEnDAR which would handle the starting and stopping of the process model. It would also require that the implemented model was able to dump the full existing model state and then be restarted with an updated version of this state (as is possible with JULES). In this iterative framework accounting for model error would also become more important.

A particularly appealing aspect of the LaVEnDAR framework as presented in this paper is that there is no interaction between the DA technique and the model itself — once the initial ensemble is generated it is not necessary to run the model again to perform any aspect of the DA. Because the main computational overhead is running the model this makes the DA *analysis* extremely efficient. This is quite unlike related techniques such as 4DVar and provides some unique opportunities. For example, it lends itself to efficient implementation of Observing System Simulation Experiments (OSSEs). OSSEs are used to examine the impact of different observation networks and sampling strategies on specific model Data Assimilation problems by repeating twin experiments with different sets of synthetic observations used to mimic different instruments and/or sampling regimes. In the LaVEnDAR framework the synthetic observations for a large number of different scenarios can all be generated with the initial ensemble and hence facilitate a large number of OSSE experiments without any further model runs.

## 5 Conclusions

Variational DA with land surface models holds a lot of potential, especially for parameter estimation, but as land surface models become more complex and subject to more frequent version releases the calculation and maintenance of a model adjoint will become increasingly expensive. One way to avoid the computation of a model adjoint is to move to ensemble data assimilation methods. In this paper we have documented the LaVEnDAR framework for the implementation of 4DEnVar data assimilation with land surface models. We have shown the application of the LaVEnDAR DA framework to the JULES land surface model, but as it requires no modification to the model itself it can easily be applied to any land surface model. Using the LaVEnDAR framework with JULES we retrieved a set of "true" model parameters given known prior and observation error statistics in a set of twin experiments and improved model predictions of real world observations from the Mead continuous maize US-Ne1 FLUXNET site. The use of 4DEnVar with land models holds a great deal of potential for both parameter and state estimation. The additional computational overhead compared to 4DVar is an appealing compromise given the simplicity and generality of its implementation.

*Code and data availability.* The code and documentation used for the experiments in this paper is available from https://github.com/pyearthsci/lavendar (last access: 20 February 2019) (Pinnington, 2019). Data for Mead site US-Ne1 available from http://fluxnet.fluxdata.org/

*Author contributions.* Ewan Pinnington, Tristan Quaife and Karina Williams designed the experiments. Amos Lawless provided advice on the data assimilation technique and tests. Ewan Pinnington developed the data assimilation code and performed the simulations. Tim
5   Arkebauer and Dave Scoby provided observations from the Mead US-Ne1 site. Ewan Pinnington prepared the manuscript with contributions from all co-authors.

*Competing interests.* The authors declare that they have no conflict of interest.

*Acknowledgements.* This work was funded by the UK Natural Environment Research Council's National Centre for Earth Observation ODA programme (NE/R000115/1). The US-Ne1, US-Ne2 and US-Ne3 AmeriFlux sites are supported by the Lawrence Berkeley National Lab
10   AmeriFlux Data Management Program and by the Carbon Sequestration Program, University of Nebraska-Lincoln Agricultural Research Division. Funding for AmeriFlux core site data was provided by the U.S. Department of Energy's Office of Science. Partial support from the Nebraska Agricultural Experiment Station with funding from the Hatch Act (Accession Number 1002649) through the USDA National Institute of Food and Agriculture is also acknowledged.

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
