# Peer review of "The Land Variational Ensemble Data Assimilation fRamework: LaVEnDAR"

_Geoscientific Model Development, 2019_

## Short Comment (SC1) · 24 Apr 2019

Astrid Kerkweg

kerkweg@uni-bonn.de

Dear authors,

In my role as Executive editor of GMD, I would like to bring to your attention our Editorial version 1.1:

http://www.geosci-model-dev.net/8/3487/2015/gmd-8-3487-2015.html

This highlights some requirements of papers published in GMD, which is also available on the GMD website in the 'Manuscript Types' section:

http://www.geoscientific-model-development.net/submission/manuscript_types.html

In particular, please note that for your paper, the following requirements have not been

met in the Discussions paper:

- "The main paper must give the model name and version number (or other unique identifier) in the title."

- "All papers must include a section, at the end of the paper, entitled 'Code availability'. Here, either instructions for obtaining the code, or the reasons why the code is not available should be clearly stated. It is preferred for the code to be uploaded as a supplement or to be made available at a data repository with an associated DOI (digital object identifier) for the exact model version described in the paper. Alternatively, for established models, there may be an existing means of accessing the code through a particular system. In this case, there must exist a means of permanently accessing the precise model version described in the paper. In some cases, authors may prefer to put models on their own website, or to act as a point of contact for obtaining the code. Given the impermanence of websites and email addresses, this is not encouraged, and authors should consider improving the availability with a more permanent arrangement. After the paper is accepted the model archive should be updated to include a link to the GMD paper."

Thus, add a version number of LaVEnDAR to the title. We very much appreciate, that the code is freely available. However, to fully meet the two criteria to archive the **exact** code version should be **permanently** archived, please provide a DOI for the exact code version published in this article. Note that for projects in GitHub a DOI can easily be created using Zenodo, see https://guides.github.com/activities/citable-code/ for details.

Yours,

Astrid Kerkweg

---

## Author Comment (AC1) · 2 May 2019

Dear Astrid Kerkweg,

Thank you for your comment and highlighting the need to attach a DOI to the version of the code presented here. We have done as you suggest and used Zenodo, the DOI is: http://doi.org/10.5281/zenodo.2654853, we will make sure this appears in the final manuscript along with the version number (v1.0.0) in the title.

Kind Regards, Ewan
* * *

---

## Referee Comment (RC1) · Anonymous Referee #1 · 12 May 2019

This is a short, neat paper describing the initial application of a 4DEnVar technique to the JULES land surface model. Results first from a twin experiment and then a test at an agricultural site are presented and seem to show considerable promise for this technique to estimate model parameters in these circumstances. It is well written, presenting the ideas clearly and concisely with very few grammatical/spelling errors and is well within the scope of GMD.

The authors articulate well the needs for such an approach – developing and maintaining an adjoint for rapidly evolving land surface models is an almost impossible task. Being model agnostic is likely to make this a potentially useful tool to many modeling groups. Their assertion that an additional benefit of the approach is that it identifies "true" parameters that are defined as being static in time seems tenuous and superfluous, not least as they also suggest in the preceding paragraph that land model parameters can, and might be expected to, change over time. This is a distraction, and it is in the nature of land surface models that so called "parameters" are not equivalent to physical constants more common in fluid dynamics models.

It is beyond this reviewer's skill set to verify the validity of all the equations in sections 3.2.1 and 3.2.2, but with my level of understanding they seem to be complete. However, given the role of this paper in introducing the new framework it would be very good to have a small amount of addition information as to how the system is actually implemented. This is most important addition this manuscript requires to increase is benefit to the community.

In addition to this, overall the manuscript would benefit from clarifications about the following points, listed in general order of appearance.

Page 9 Line 12 What does "2% Gaussian noise" mean? 1 standard deviation = 2% of value at time of sampling. And presumably this is also used for the observation error variance?

P10L2 Why do you use different variances for the parameter perturbations in the twin v. real experiment? Would it not have been more informative to use the same (ie higher) value for the twin?

P10L3 The justification for arbitrarily assigning observation errors is insufficient. How do these compare to actual estimates from field studies. In particular, using a percentage value for the fluxes is a poor choice – absolute errors don't scale well with flux magnitude. Again, using different/unknown values for the twin experiment relative to the real data is not helpful when making comparisons between the two.

P10L17 The authors comment multiple times on the error introduced/reduced around due variability in harvesting dates. That in itself may be a distraction, but nevertheless if it important the model description needs to include details as to how its calculated.

P10L20 Similarly, how is harvestable material calculated? This is particularly important to know as later there is an apparent discrepancy between good harvestable material estimates, but poor leaf C and stem C estimates.

P11L3 The parameter priors don't seem to be very normally distributed, but they should be? Mu in particular is seems weighted around the true value?

P15L7 "only capturing 5 of the 11 observations" What is meant by this? It seems like all but one observation lie within +- 1 SD in the prior case, although that is wide and poorly constrained. What metric is being used?

P15L15 If LAI agrees with observations, but leaf C does not, this implies SLA is incorrect, but this is one of the parameters being optimized, or at least a coefficient controlling it? What is the suggestion of this for the model?

P20L2 How will the correlations in the prior error covariance matrix be determined/estimated?

P20L4 To what extent is this ensemble collapse a function of (over optimistic) observation error?

P20L28 Again, how is harvestable material calculated? It would appear they might be compensatory biases given harvestable material estimates seem better than either leaf C or stem C

P21L4 A brief discussion of the steps required to extend this framework to models running on spatial grid regionally/globally in addition to a need for localization would be very beneficial, including any potential limitations.

P21L9 Can you elaborate on how you intend to use this framework in a cycling system? Over what sort of timescales would you run the model before restarting. Won't this result in variation in the "true parameters"?

P21L20 This sentence is unclear and need editing.

---

## Referee Comment (RC2) · Anonymous Referee #2 · 14 Jun 2019

In this paper Pinnington et al introduce the land modeling community to 4D ensemble variational data assimilation (4DEnVar) and demonstrate its application to a land model , JULES, at a site-scale. This system is applied in two examples, one using simulated data with known parameters, the other using field data, to demonstrate the ability of 4DEnVar to constrain model parameters. The system appears to be updating initial conditions as well, though these were not discussed. The Python code base to support these two examples is given the name LaVEnDAR, and while the approach is touted as being 'general', it is not entirely clear whether the system has been applied to other sites, models, or data constraints, nor is it clear what would be required to be able to do so. That said, it is fully worth acknowledging that the work presented in this paper is definitely sufficient to constitute a 'first application' of the LaVEnDAR system and

that the system appears to perform well. In the detailed comments below I will raise a number of concerns about the current capabilities of this system, but I want to be clear at the outset that I do not consider these to be fatal flaws for this initial paper, but rather important caveats that need to be stated more explicitly in the paper and then improved upon in subsequent applications.

Page 1, Lines 15-17: Both your land surface model and atmospheric models are completely deterministic. The chaotic behavior of the atmosphere is indicative of its high sensitivity to initial conditions, not stochasticity. The important difference I think you're trying to get at, which I'll agree has very important implications for prediction and data assimilation, is that JULES is a stable model that will converge to a steady state.

P1, L19: (i) "problem _of_ parameter estimation" (ii) I don't agree that it is safe to assume that parameter estimation is the only issue here, or even that parameter uncertainty is the dominant uncertainty in land model prediction. I don't think this has really been shown conclusively, as all existing uncertainty analyses I'm aware of either ignore or confound multiple key uncertainties. I agree calibration is definitely an important problem, but don't oversell/overstate your position.

P2, L9: "parameters can change over time" – this point is debatable. I'd say that it's probably more appropriate to say that allowing model parameters to change over time (or space) is a mechanism that can be used to account for model structural inadequacy (processes or covariates that are missing from the model). Of course that inadequacy/incompleteness is an inherent feature all models, so to some degree parameter variability (typically modeled using random effects) is frequently a source of uncertainty that needs to be considered.

P2, L12: If the focus is on efficient approached to model calibration, I'd recommend mentioning emulator methods as well (e.g. Fer et al 2018 Biogeosciences)

P2, L14: I don't think "non-Gaussiantity" is a word. Maybe "non-Gaussian error" instead?

P2, L21: I'm surprised the paper is adopting the position that parameters should be static in time after arguing just 12 lines ago that parameters change over time.

P3 L31: GPP is not an observation, it is predicted from a simple model based on NEE and environmental covariates, and those simple models are known to have errors and systematic biases. Treating GPP like it is data means you are calibrating your model to another model, which should be treated with extreme caution.

P4 L14-15: I find this notation to be unnecessarily confusing. Specifically, why use i as a subscript instead of t when i is being used to indicate time? It'd be much simpler to just use t and t-1. Also, do we really need a subscript on f? Is the model itself changing with time?

P4, L25: won't this structure change the time invariance of p once you account for process error (unless you define the variance as zero, but that'll probably mess up the inversion of the error covariance matrix)

P6 L6: This bit is really in the weeds and could benefit from a bit more detail/explanation.

P7 L17: Here you say the adjoint is still present, but this is the first mention of an adjoint in the Methods. Needs further explanation.

Figure 1: I'm not sure this figure is useful. I'd either drop or combine it with Figure 2

P9, L6: Why these seven parameters? Were there any sort of uncertainty analyses performed that attributed model uncertainty to these parameters specifically?

P9, L11: Why was this variance chosen? Does this represent a typical or realistic level of parameter uncertainty? My experience has been that the magnitude of parameter variance can differ enormously from parameter to parameter because of the wildly differing amounts of trait data available to constrain different parameters.

P9, L12: This is definitely an unrealistically low amount of noise on any sort of land

observations, especially in light of the fact that process error is not included. It might be useful to develop some additional analyses that explore larger, more realistic observation errors.

P10, L2-4: Choice of variances here (initial conditions and observation errors) are similarly not given any sort of justification and strike me as much too tight.

P19, L8: But this raises the question about why this parameter was selected for inclusion in the calibration, out of the 90 PFT-level parameters in JULES http://jules-lsm.github.io/vn4.9/namelists/pft_params.nml.html, if model outputs are not sensitive to it.

P20, L4-14: I find it odd that this paragraph discusses the ability of 4DEnVar to accurately retrieve parameters as if it were a bad thing. At the heart of the issue is the (unstated) problem of filter divergence, where the model ensemble becomes sufficiently confident in itself that it ignores (diverges from) the observations. The authors' concern here suggests a misrepresentation of the uncertainties that control the ensemble spread. Specifically, of the five uncertainties that control the spread of the ensemble (initial conditions, external drivers, parameter uncertainty, parameter variability [i.e. random effects], and process error; see Dietze 2017 Ecol Appl), the current analysis is only considering two (IC and parameter uncertainty). Because, as discussed at P1 L15, the model is stable the IC uncertainty will decline exponentially toward zero with time. Similarly, parameter uncertainty will decline asymptotically toward zero with more data (and since the data are timeseries, that implies that this uncertainty also declines with time). So it is unsurprising that the ensemble is converging toward zero variance, as that's exactly what we know it should do from first principles. By contrast, the three uncertainties not included in the current analysis (drivers, random effects, process error) all systematically increase the ensemble variance with time. There are also ∼80 other PFT-level parameters in JULES whose (prior) uncertainty isn't being propagated. Rather than suggesting the use of methods to inflate the ensemble variance, I'd argue that the authors would be much better served by including the missing uncertainties

that do this naturally. As discussed in my overall summary, I'm not asking the authors to redo their current analysis, but I am asking that they revise their Discussion to acknowledge the future steps that need to be taken to incorporate these missing uncertainties.

P20, L8: If parameters trading off is an issue, I'd recommend reporting the posterior covariances (or correlations) as a supplement. You fundamentally can't see equifinality in the parameter means or posterior marginal distributions, as it is a property of the JOINT posterior distribution.

P20, L13: While I previously suggested that you drop ensemble inflation methods altogether from the Discussion, if you do retain this I'll note that this statement is not explained sufficiently to the reader.

P21, L3: Either explain what ensemble localization is, or drop.

P21, L17: As noted P1, L19 I don't think this issue is settled. It's worth noting that the paper cited as justification likewise only considers a subset of the uncertainties mentioned in my comment on P20, L4-14

P21, L22: (i) as noted in P20, L4-14 comment, don't equate process error with "stochastic noise" or inflation. (ii) It is also worth noting that data assimilation frameworks are usually applied iteratively, but the current proof-of-concept application of LaVENDAR is a completely 'offline' problem that's never applied to more than a year's worth of data. I think this requires more discussion and acknowledgement of the current system's limitations. There are a number of additional modules that would need to be added to LAVENDAR to support this, as well as features that need to be present in the model itself. Specifically, while LAVENDAR claims to be able to work on any existing model based just on it's outputs, iteration would require the ability to save the full state of the model and then restart the model from updated initial conditions. I'd recommend expanding this discussion. Similarly, by only applying the 4DEnVar to one year of data (neither assimilating a second year nor validating against a second year)

the authors were able to skip over the problem that using fixed parameters will most likely leave the model unable to capture interannual variability. The single-year application may thus be leaving readers with an overly-optimistic view of how well the system is performing. I'd recommend discussing this explicitly or (even better) demonstrating it with additional validation years.

―――――――――――――――――――

---

## Author Comment (AC2) · 9 Jul 2019

We thank the reviewer for their comments which have helped strengthen the manuscript. Please find attached a zip file containing a pdf of our responses, a pdf of an updated manuscript, a pdf of the latexdiff between update and original manuscript with proposed changes highlighted in blue and deletions in red and a pdf of additional supplementary material requested by the reviewers.

The response document is structured with reviewer's comments in black text and author responses in blue text.

Kind Regards, Ewan Pinnington

[Figure]

Please also note the supplement to this comment:
https://www.geosci-model-dev-discuss.net/gmd-2019-60/gmd-2019-60-AC2-supplement.zip

---

## Author Response (AR1)

LaVEnDAR response to reviewers

We thank the reviewers for their careful attention to detail on this manuscript. Their comments have undoubtedly helped strengthen the paper.

**RC1:**
"Their assertion that an additional benefit of the approach is that it identifies "true" parameters [...]". The reviewer suggested that our assertion in the introduction that an additional benefit of the technique is that it identifies "true" parameters which are static in time was tenuous and superfluous

Our brief comment on time varying parameters was perhaps not well thought through - and this was picked up by both reviewers. Our intention was to distinguish between the result we would have obtained using a filter rather than a variational technique. However, we agree with R#1 that this was superfluous. To improve the clarity of the manuscript we have removed these sentences.

"[...] it would be very good to have a small amount of addition information as to how the system is actually implemented". The reviewers main request was for the addition of information on the implementation of the technique

We agree with the reviewer that this was an obvious omission in the manuscript and have added a new section describing the implementation in JULES and what would be required for a more general implementation in other models on page 8 at line 5.

"In order to implement 4DEnVar we construct an ensemble of parameter vectors and then run the process model for each unique parameter vector over some predetermined time window. We then extract the ensemble of model-predicted observations from the ensemble of model runs and compare these with the observations to be assimilated over the given time window. In our code (Pinnington, 2019) we implement the method of 4DEnVar with JULES using a set of Python modules. The data assimilation routines and minimization are included in fourdenvar.py. This part of the code does not need to be modified to be used with a new model. Model specific routines for running JULES are found in jules.py and run_jules.py. JULES is written in FORTRAN with its parameters being set by FORTRAN namelist (NML) files; jules.py and run_jules.py operate on these NML files updating the parameters chosen for optimisation. The data assimilation experiment is setup in experiment_setup.py with variables set for output directories, model parameters, ensemble size and functions to extract observations for assimilation. The module run_experiment.py runs the ensemble of models and executes the experiment as defined by experiment_setup.py. Some experiment specific plotting routines are also included in plot.py. More information and a tutorial can be found at https://github.com/pyearthsci/lavendar.

To use another model in this framework new wrappers would have to be written to mimic the functionality of jules.py and run_jules.py and allow for multiple model runs to be conducted while varying parameters. The module run_experiment.py would need to be updated to account for these new wrappers and functions to extract the observations for assimilation included in experiment_setup.py. Although we have used Python here to implement a stand-alone setup of LaVEnDAR we envisage that the technique could be added to existing

workflow systems such as Cylc (Oliver et al., 2019) or the Predictive Ecosystem Analyzer (PEcAn) (LeBauer et al., 2013)."

P9 L12: What does 2% Gaussian noise mean?
We have reworded this at page 10 line 4.

"Perturbed using Gaussian noise with a standard deviation of 2% of the synthetic truth value."

P10L2 The reviewer asked why we chose to have lower uncertainties in the twin vs. the real experiment and thought it might be more informative to use the same uncertainties for the twin as in the real experiment.
The purpose of the twin-experiments is to demonstrate that we can retrieve correct parameters when we have high confidence on the observations and priors. When observations and priors are less certain (as is the case in the real-world experiment) retrieving the "true" parameter values is not guaranteed and hence is a less clear test of the data assimilation system. However, we agree that using the same uncertainties can also be informative and have now included the suggested experiment in supplementary material and added reference in the main text at page 10 line 7.

"We also include a twin experiment using the same error statistics as those used for the real data experiments at the Mead site (outlined in section 2.4.2) in supplementary material section S1.1."

P10L3 The reviewer commented that our justification of observational errors, in particular GPP, was insufficient
We agree that our justification of errors in GPP data is simplistic, and for the purpose of making robust scientific inference a greater level of effort would be required to quantify the uncertainty in the flux data. We have added references to the text demonstrating that a 10% error on GPP could be a reasonable choice on page 10 line 17.

"We prescribe a 5% standard deviation for canopy height and leaf area index errors and a 10% standard deviation for errors in GPP. These uncertainties are rough estimates that we considered adequate for demonstrating our system, but for any specific application the errors estimates should be determined more carefully. However, our uncertainties are consistent with Schaefer et al. (2012) who found an uncertainty of 1.04 g C m-2 day-1 to 4.15 g C m-2 day-1 (scaling with flux magnitude) for estimates of GPP, Raj et al. (2016) who found an uncertainty in the order of 10% for daily estimates of GPP, and Guindin-Garcia et al. (2012) who found a standard error of 0.15 m2 m-2 for destructively sampled green LAI at the Mead flux site."

P10L17/P10L20/P20L28 The reviewer asked for the inclusion of a description of how model harvest date and harvestable material are calculated
We have now included a brief description of how harvest date and harvestable material is calculated in the model and cited a paper that provides details of the complete algorithm at page 3 line 25.

"Crop development is governed by a crop development index which increases as a function of crop-specific thermal time parameters with the crop being harvested when the development index crosses certain thresholds. The crop grows by accumulating daily NPP and partitioning this between a set of carbon pools (havestable material, leaf, root, stem and reserve), equations for JULES-crop can be found in Williams et al. (2017) appendix A1."

P11L3 The reviewer commented that some of the parameter priors (particularly mu) did not seem to be very normally distributed.
Here we are only sampling 50 ensemble members for the prior and then fitting a curve to the parameter histogram. We are also sampling a bounded distribution to ensure none of the sampled prior parameters are below zero (except for delta which is negative). Due to the small number of samples and the boundedness the prior distributions sometimes don't appear very Gaussian.

P15L7 Reviewer asked what was meant by "only capturing 5 of the 11 observations"
Here we were referring to how many observations the ensemble mean passed through, rather than how many observations were captured by the ensemble spread. We agree this was not at all clear and have updated the wording accordingly on page 15 line 6.

"From Figure 7 we can see that the prior mean underestimates LAI, reaching a much lower peak than observations, despite this the technique finds a posterior mean estimate that agrees well with all but 2 LAI observations (in September and October)."

P15L15 If LAI agrees with observations, but leaf C does not, this implies SLA is incorrect, but this is one of the parameters being optimized, or at least a coefficient controlling it? What is the suggestion of this for the model?
It is likely that the optimized parameters controlling SLA are compensating for error in the parameters controlling the partitioning of NPP into the leaf carbon pool. This allows us to achieve the correct leaf area with the incorrect leaf carbon content. We have added discussion on this at page 15 line 16.

"The fact that we can find good agreement for LAI with a poorer fit to leaf carbon content is likely due to the optimised parameters controlling specific leaf area compensating for errors in model parameters controlling the partitioning of net primary productivity into the leaf carbon pool. This allows us to achieve the correct leaf area but with the incorrect leaf carbon content."

P20L2 How will the correlations in the prior error covariance matrix be determined/ estimated?
For the purposes of testing the 4DEnVar technique we did not consider error covariances in our prior information. We have published on this topic previously and include a brief discussion of this at page 20 line 3 in the current manuscript.

"Alternatively including correlations in the prior error covariance matrix would provide information to update *fd* even when the assimilated observations are not impacted by changes in this parameter. It has been shown that suitable correlations can be diagnosed by

sampling from a set of predetermined ecological dynamical constraints and taking the covariance of an ensemble run forward over a set time window (Pinnington et al. 2016)."

P20L4 To what extent is this ensemble collapse a function of (over optimistic) observation error?
Ensemble collapse here was a poor choice of phrasing - we did not mean it in the traditional sense used in data assimilation. We meant only that the ensemble had converged around a particular value rather than "collapsed" which is normally used to indicate all ensembles occupying the same space. We have modified the text at page 20 line 6.

P21L4 A brief discussion of the steps required to extend this framework to models running on spatial grid regionally/globally in addition to a need for localization would be very beneficial, including any potential limitations.
We have added this discussion on page 21 line 18

"In order to extend this framework to model runs over a spatial grid we will need a method to sample prior parameter distributions regionally or globally, it would then be possible to conduct parameter estimation experiments over a region, either on a point by point basis or for the whole area at once. Considering a large area would increase the parameter space and require more ensemble members. Localisation in space could help to reduce the number of ensemble members required."

P21L9 The reviewer asked how this framework would be used in a cycling system
The use of a cycling system is probably more appropriate to state assimilation rather parameter estimation. Once parameter estimation had been conducted the framework could be set up for state estimation and then cycled on a timescale suitable for the desired target variable and data availability. We have elaborated on this in the text page 21 line 27.

"While posterior parameter estimates could be used in future studies with their associated uncertainties we envisage that cycling of the assimilation system will be more appropriate for state estimation (after initial parameter estimation) where the system could be cycled on a timescale suitable for the required state variable and data availability."

P21L20
Typo sentence modified.

LaVEnDAR response to reviewers

We thank the reviewers for their careful attention to detail on this manuscript. Their comments have undoubtedly helped strengthen the paper.

**RC2:**
The reviewer commented that it was not clear if this was the first application of LaVEnDAR or not and queried what would be needed to apply this technique to other problems. This is indeed the first application of LaVEnDAR. We have added text to the abstract and introduction to make this clear page 1 line 4

"In this paper we present the first application of LaVEnDAR, implementing the framework with the JULES land surface model."

and page 2 line 33.

"In this paper we present the first application of the Land Variational Ensemble Data Assimilation fRamework (LaVEnDAR) for implementing the hybrid technique of Four-Dimensional Ensemble Variational Data Assimilation (4DEnVar) with land surface models."

We have also added a new section on the implementation of LaVEnDAR including which modules would need to be changed for application to another problem on page 8 and line 5.

"In order to implement 4DEnVar we construct an ensemble of parameter vectors and then run the process model for each unique parameter vector over some predetermined time window. We then extract the ensemble of model-predicted observations from the ensemble of model runs and compare these with the observations to be assimilated over the given time window. In our code (Pinnington, 2019) we implement the method of 4DEnVar with JULES using a set of Python modules. The data assimilation routines and minimization are included in fourdenvar.py. This part of the code does not need to be modified to be used with a new model.  Model specific routines for running JULES are found in jules.py and run_jules.py. JULES is written in FORTRAN with its parameters being set by FORTRAN namelist (NML) files; jules.py and run_jules.py operate on these NML files updating the parameters chosen for optimisation. The data assimilation experiment is setup in experiment_setup.py with variables set for output directories, model parameters, ensemble size and functions to extract observations for assimilation. The module run_experiment.py runs the ensemble of models and executes the experiment as defined by experiment_setup.py. Some experiment specific plotting routines are also included in plot.py. More information and a tutorial can be found at https://github.com/pyearthsci/lavendar.

To use another model in this framework new wrappers would have to be written to mimic the functionality of jules.py and run_jules.py and allow for multiple model runs to be conducted while varying parameters. The module run_experiment.py would need to be updated to account for these new wrappers and functions to extract the observations for assimilation included in experiment_setup.py. Although we have used Python here to implement a

stand-alone setup of LaVEnDAR we envisage that the technique could be added to existing workflow systems such as Cylc (Oliver et al., 2019) or the Predictive Ecosystem Analyzer (PEcAn) (LeBauer et al., 2013)."

P1L15-17: The reviewer pointed out that both land surface and atmospheric models are deterministic.
We agree that our description of land surface and atmospheric models here is incorrect and have updated the text accordingly at page 1 and line 15.

"Most land surface models will converge to a steady state; their state vector tends toward an equilibrium defined by forcing variables (i.e. the meteorology experienced by the model) and the model parameters. This is quite unlike fluid dynamics models used for the atmosphere and oceans, which exhibit chaotic behaviour; a small change in their initial state can lead to large deviations in the state vector evolution with time."

P1L19: The reviewer pointed out a typo and thought we were overstating the problem of parameter estimation.
We have corrected the typo and moderated our statement of the problem on page 1 and line 18.
"Consequently, for some land surface applications parameter estimation can have greater utility than state estimation. This manuscript deals primarily with the problem of parameter estimation in land surface models, although the technique we introduce could easily be used to for state estimation problems too."

P2L9: The reviewer suggested that allowing parameters to change in time was a way of accounting for model structural inadequacies.
We have modified the text to reflect this at page 2 and line 8.

"However, this is not true for land surface models where parameters are much less well understood. Indeed these parameters can be allowed to change over time within a developing ecosystem or when an ecosystem is subject to a disturbance event to account for model structural inadequacies."

P2L12: The reviewer thought it was worth mentioning emulator methods here also.
We have added comment on these methods as requested at page 2 line 22

"There is also a growing interest in model emulation, (Gómez-Dans et al., 2016; Fer et al., 2018), these techniques are extremely efficient but require some initial construction of the emulator."

P2L14: non-Gaussianity not a word maybe "non-Gaussian error" instead?
Updated.

P2L21: I'm surprised the paper is adopting the position that parameters should be static in time after arguing just 12 lines ago that parameters change over time.

Our intention was to argue *against* time varying parameters and we obviously did not succeed in that very clearly as Reviewer 1 also commented on this. As described in our response to R#1 comment 1 we have deleted the text around this as it caused confusion and, ultimately, did not motivate the development of the DA tool we have presented.

P3L31: The reviewer commented that GPP is not an observation and using this data in the assimilation should be treated with extreme caution.
We agree with this and have added text at page 4 line 5 to add caution.

"It is important to note that GPP is not an observation *per se* and is derived by partitioning the net carbon flux using a model which is likely to be inconsistent with the process model we are assimilating the data into."

P4L14-15: The reviewer thought our notation was confusing here and suggest we change the i subscript to a t. They also asked if we need a subscript on the model, *f,* and if this represented the model changing in time.
We agree that just using a t subscript may make things clearer for the reader, we have made this change throughout the manuscript. The subscript on f is not representing the model changing in time but repeated applications of the model to update the state to the desired time step. This then forms the basis for the matrix notation in equation (13).

P4L25: The reviewer asked if this structure would change the time invariance on p when accounting for process error.
It is possible to set up the assimilation system to include process error, but it has not been done in this case. Equation 5 deals only with the formation of the augmented state-vector. The reviewer is correct however that this point in the system is where we would add the process error if required. This would result in variation in p with time but it would be possible to prescribe a small variance in the process error to keep the change in p minimal, if this was the desired behaviour. We have added text to the paper discussing this at page 5 line 3.

"Process error could be included in equation (5) by specifying an additional term, but in this application is neglected."

P6 L6: The reviewer commented that more detail would be beneficial here.
We agree extra description is helpful here and have included this at page 6 line 11.

"For certain applications the prior error covariance matrix **B** can become large, ill-conditioned and difficult to invert. As a result minimising the cost function in equation (11) and finding the optimised model state/parameters can be slow. To ensure the 4DVar cost function converges as efficiently as possible and to avoid the explicit computation of the matrix **B** the problem is often preconditioned using a control variable transform (Bannister, 2016). We define the preconditioning matrix **U** by,"

P7L17: Here you say the adjoint is still present, but this is the first mention of an adjoint in the Methods. Needs further explanation.
We have added description of the adjoint earlier in the methods section page 5 line 24.

"**M**$^T_{t,0}$ is the model adjoint propagating the state backward in time (this is required for efficient minimisation of the cost function using gradient descent techniques)."

Figure 1: The reviewer thought this figure was not helpful.
We have removed this figure.

P9L6: The reviewer wanted to know why we picked the parameters we did for the experiments and if there was uncertainty analysis conducted that attributed model uncertainty to these specific parameters.
We have included a sentence on this in the text page 9 line 13. See also response to comment P9L11/P10/L2-4.

"These seven parameters have an effect on the crop's seasonal growth cycle and its photosynthetic response to meteorological forcing data. The choice of parameters was motivated by the analysis of Williams et al. (2017) who found that they were least able to constrain these parameters with the available data"
Williams, K., Gornall, J., Harper, A., Wiltshire, A., Hemming, D., Quaife, T., Arkebauer, T., and Scoby, D.: Evaluation of JULES-crop performance against site observations of irrigated maize from Mead, Nebraska, Geosci. Model Dev., 10, 1291-1320, https://doi.org/10.5194/gmd-10-1291-2017, 2017.

P9L11/P10L2-4: The reviewer asked what the reasoning was behind the choice of parameter variance values for both twin and Mead experiments and that the priors and observation errors be given some justification.
We agree that a more rigorous approach could be taken to assigning the parameter uncertainties. As the analysis of Williams et al. (2017) showed all parameters to be poorly constrained with available data in a more traditional model calibration study we applied a blanket variance to all parameters. Reviewer 1 also asked for justification of observation errors. We have included extra text on this at page 10 line 13.

"We apply the same variance to all parameters here as the analysis of Williams et al. (2017) showed these parameters to all be poorly constrained with the available data in a more traditional model calibration study. In reality it is unlikely that all parameters will have the same variance but in the absence of additional information and for the purposes of this demonstration we used $(0.25 \times x_b)^2$ […] We prescribe a 5% standard deviation for canopy height and leaf area index errors and a 10% standard deviation for errors in GPP. These uncertainties are rough estimates that we considered adequate for demonstrating our system, but for any specific application the errors estimates should be determined more carefully. However, our uncertainties are consistent with Schaefer et al. (2012) who found an uncertainty of 1.04 g C m-2 day-1 to 4.15 g C m-2 day-1 (scaling with flux magnitude) for estimates of GPP, Raj et al. (2016) who found an uncertainty in the order of 10% for daily estimates of GPP and Guindin-Garcia et al. (2012) who found a standard error of 0.15 m2 m-2 for destructively sampled green LAI at the Mead flux site."

P9 L12: The reviewer stated that the observational noise was much too low in the twin experiments and that it would be beneficial to repeat the twin experiment with large uncertainties.

Reviewer 1 had a similar comment that we should repeat the twin experiments using the same error statistics as in the real-world Mead experiment, for which we replied:

The purpose of the twin-experiments is to demonstrate that we can retrieve correct parameters when we have high confidence on the observations and priors. When observations and priors are less certain (as is the case in the real-world experiment) retrieving the "true" parameter values is not guaranteed and hence is a less clear test of the data assimilation system. However, we agree that using the same uncertainties can also be informative and have now included the suggested experiment in supplementary material and added reference in the main text at page 10 line 7.

"We also include a twin experiment using the same error statistics as those used for the real data experiments at the Mead site (outlined in section 2.4.2) in supplementary material section S1.1."

P19L8: The reviewer again queried the choice of parameters asking why fd was selected if model outputs were not sensitive to it.

See comment P9L6. Although we have not been able to recover this parameter we believe this is a good example of an instance where unobservable parameter/state members become a problem.

P20 L4-14: The reviewer asked us to discuss the other sources of uncertainty not included in this study and how these might be included in the 4DEnVar framework.

We have included a discussion of the other sources of error noted by the reviewer and how these might be included in the DA system at page 20 line 16.

"In this study we have only considered the uncertainty in the parameters and initial conditions and not the uncertainty in forcing data, random effects (parameter variability) or uncertainty in the process model (Dietze, 2017). The inclusion of these additional sources of error would avoid the ensemble converging too tightly around any given value. In order to include uncertainty in the forcing data it would be necessary to run each ensemble member with a different realisation of the driving meteorology. Process error could be included in equation (5) resulting in a new term in the 4DEnVar cost function in equation (24) containing a model error covariance matrix, it has also been shown that these different types of uncertainty could be built into the observation error covariance matrix R (Howes et al., 2017). If estimates to these sources of error are not available the use of methods such as ensemble inflation (Anderson and Anderson, 1999), a set of techniques where the ensemble spread is artificially inflated, will help alleviate problems of ensemble convergence."

P20L8: The reviewer asked us to include posterior covariances (or correlations) as a supplement to this discussion.

We have included the posterior correlation matrix in supplementary material and referenced this at page 20 line 11.

"From table 3 we can see this issue for the two parameters controlling photosynthetic response with the posterior slightly over-predicting α and under-predicting *neff*, as different

combinations of these parameters can produce the same trajectory for the observed target variables. The effect of equifinality can be seen more clearly for the posterior ensemble correlation matrix included in Figure S7 of the supplementary material."

P20L13: The reviewer suggested we had not explained ensemble inflation adequately.
We have expanded on this in the text, see comment P20L4-14.

P21L3: The reviewer asked us to include a description of localization or drop it.
We have included a brief description of localization at page 21 line 16

"Methods of ensemble localisation (Hamill et al., 2001), where distant correlations or ensemble members are down-weighted or removed, could be used to improve prior estimates."

P21L17: "[...] parameter estimation as it is often more important [...]". The reviewer commented that the matter is not settled.
We have removed this sentence.

P21L22: (i) The reviewer noted that we should not equate process error with stochastic noise or inflation (ii) The reviewer asked for more discussion of how iteration would work in LaVEnDAR and that we consider adding an additional year for validation.
(i) We have removed discussion of stochastic noise and inflation here. (ii) We have added more discussion on the iteration at page 22 line 6

"This would require additional modules to be written within LaVEnDAR which would handle the starting and stopping of the process model. It would also require that the implemented model was able to dump the full existing model state and then be restarted with an updated version of this state (as is possible in JULES). In this iterative framework accounting for model error would also become more important."

and also included an additional year of validation (i.e. a hindcast) in supplementary material, where we have run our posterior ensemble for 2008 across 2009. We have added a reference to this at page 20 and line 34

[revised manuscript text omitted]